# A PAC-Bayesian Analysis of Randomized Learning with Application to Stochastic Gradient Descent

**Ben London**
blondon@amazon.com
Amazon AI

## Abstract

We study the generalization error of randomized learning algorithms—focusing on stochastic gradient descent (SGD)—using a novel combination of PAC-Bayes and algorithmic stability. Importantly, our generalization bounds hold for all posterior distributions on an algorithm's random hyperparameters, including distributions that depend on the training data. This inspires an adaptive sampling algorithm for SGD that optimizes the posterior at runtime. We analyze this algorithm in the context of our generalization bounds and evaluate it on a benchmark dataset. Our experiments demonstrate that adaptive sampling can reduce empirical risk faster than uniform sampling while also improving out-of-sample accuracy.

## 1 Introduction

Randomized algorithms are the workhorses of modern machine learning. One such algorithm is stochastic gradient descent (SGD), a first-order optimization method that approximates the gradient of the learning objective by a random point estimate, thereby making it efficient for large datasets. Recent interest in studying the generalization properties of SGD has led to several breakthroughs. Notably, Hardt et al. [10] showed that SGD is stable with respect to small perturbations of the training data, which let them bound the risk of a learned model. Related studies followed thereafter [13, 16]. Simultaneously, Lin and Rosasco [15] derived risk bounds that show that early stopping acts as a regularizer in multi-pass SGD (echoing studies of incremental gradient descent [19]).

In this paper, we study generalization in randomized learning, with SGD as a motivating example. Using a novel analysis that combines PAC-Bayes with algorithmic stability (reminiscent of [17]), we prove new generalization bounds for randomized learning algorithms, which apply to SGD under various assumptions on the loss function and optimization objective. Our bounds improve on related work in two important ways. While some previous bounds for SGD [1, 10, 13, 16] hold in expectation over draws of the training data, our bounds hold with high probability. Further, existing generalization bounds for randomized learning [6, 7] only apply to algorithms with fixed distributions (such as SGD with uniform sampling); thanks to our PAC-Bayesian treatment, our bounds hold for all posterior distributions, meaning they support data-dependent randomization. The penalty for overfitting the posterior to the data is captured by the posterior's divergence from a fixed prior.

Our generalization bounds suggest a sampling strategy for SGD that adapts to the training data and model, focusing on useful examples while staying close to a uniform prior. We therefore propose an adaptive sampling algorithm that dynamically updates its distribution using multiplicative weight updates (similar to boosting [8, 21], focused online learning [22] and exponentiated gradient dual coordinate ascent [4]). The algorithm requires minimal tuning and works with any stochastic gradient update rule. We analyze the divergence of the adaptive posterior and conduct experiments on a benchmark dataset, using several combinations of update rule and sampling utility function. Our experiments demonstrate that adaptive sampling can reduce empirical risk faster than uniform sampling while also improving out-of-sample accuracy.

## 2 Preliminaries

Let $\mathcal{X}$ denote a compact domain; let $\mathcal{Y}$ denote a set of labels; and let $\mathcal{Z} \triangleq \mathcal{X} \times \mathcal{Y}$ denote their Cartesian product. We assume there exists an unknown, fixed distribution, $\mathbb{D}$, supported on $\mathcal{Z}$. Given a dataset of *examples*, $S \triangleq (z_1, \ldots, z_n) = ((x_1, y_1), \ldots, (x_n, y_n))$, drawn independently and identically from $\mathbb{D}$, we wish to learn the parameters of a predictive model, $\mathcal{X} \mapsto \mathcal{Y}$, from a class of *hypotheses*, $\mathcal{H}$, which we assume is a subset of Euclidean space. We have access to a deterministic *learning algorithm*, $A : \mathcal{Z}^n \times \Theta \to \mathcal{H}$, which, given $S$, and some *hyperparameters*, $\theta \in \Theta$, produces a hypothesis, $h \in \mathcal{H}$.

We measure the quality of a hypothesis using a *loss* function, $L : \mathcal{H} \times \mathcal{Z} \to [0, M]$, which we assume is $M$-bounded[1] and $\lambda$-Lipschitz (see Appendix A for the definition). Let $L(A(S, \theta), z)$ denote the loss of a hypothesis that was output by $A(S, \theta)$ when applied to example $z$. Ultimately, we want the learning algorithm to have low expected loss on a random example; i.e., low *risk*, denoted $R(S, \theta) \triangleq \mathbb{E}_{z \sim \mathbb{D}}[L(A(S, \theta), z)]$. (The learning algorithm should always be clear from context.) Since this expectation cannot be computed, we approximate it by the average loss on the training data; i.e., the *empirical risk*, $\hat{R}(S, \theta) \triangleq \frac{1}{n} \sum_{i=1}^{n} L(A(S, \theta), z_i)$, which is what most learning algorithms attempt to minimize. By bounding the difference of the two, $G(S, \theta) \triangleq R(S, \theta) - \hat{R}(S, \theta)$, which we refer to as the *generalization error*, we obtain an upper bound on $R(S, \theta)$.

Throughout this document, we will view a randomized learning algorithm as a deterministic learning algorithm whose hyperparameters are randomized. For instance, *stochastic gradient descent* (SGD) performs a sequence of hypothesis updates, for $t = 1, \ldots, T$, of the form

$$h_t \leftarrow U_t(h_{t-1}, z_{i_t}) \triangleq h_{t-1} - \eta_t \nabla F(h_{t-1}, z_{i_t}),$$

using a sequence of random example indices, $\theta = (i_1, \ldots, i_T)$, sampled according to a distribution, $\mathbb{P}$, on $\Theta = \{1, \ldots, n\}^T$. The *objective* function, $F : \mathcal{H} \times \mathcal{Z} \to \mathbb{R}_+$, may be different from $L$; it is usually chosen as an optimizable upper bound on $L$, and need not be bounded. The parameter $\eta_t$ is a step size for the update at iteration $t$. SGD can be viewed as taking a dataset, $S$, drawing $\theta \sim \mathbb{P}$, then running a deterministic algorithm, $A(S, \theta)$, which executes the sequence of hypothesis updates.

Since learning is randomized, we will deal with the expected loss over draws of random hyperparameters. We therefore overload the above notation for a distribution, $\mathbb{P}$, on the hyperparameter space, $\Theta$; let $R(S, \mathbb{P}) \triangleq \mathbb{E}_{\theta \sim \mathbb{P}}[R(S, \theta)]$, $\hat{R}(S, \mathbb{P}) \triangleq \mathbb{E}_{\theta \sim \mathbb{P}}[\hat{R}(S, \theta)]$, and $G(S, \mathbb{P}) \triangleq R(S, \mathbb{P}) - \hat{R}(S, \mathbb{P})$.

### 2.1 Relationship to PAC-Bayes

Conditioned on the training data, a *posterior* distribution, $\mathbb{Q}$, on the hyperparameter space, $\Theta$, induces a distribution on the hypothesis space, $\mathcal{H}$. If we ignore the learning algorithm altogether and think of $\mathbb{Q}$ as a distribution on $\mathcal{H}$ directly, then $\mathbb{E}_{h \sim \mathbb{Q}}[L(h, z)]$ is the *Gibbs* loss; that is, the expected loss of a random hypothesis. The Gibbs loss has been studied extensively using *PAC-Bayesian* analysis (also known simply as *PAC-Bayes*) [3, 9, 14, 18, 20]. In the PAC-Bayesian learning framework, we fix a *prior* distribution, $\mathbb{P}$, then receive some training data, $S \sim \mathbb{D}^n$, and learn a posterior distribution, $\mathbb{Q}$. PAC-Bayesian bounds frame the generalization error, $G(S, \mathbb{Q})$, as a function of the posterior's divergence from the prior, which penalizes overfitting the posterior to the training data.

In Section 4, we derive new upper bounds on $G(S, \mathbb{Q})$ using a novel PAC-Bayesian treatment. While traditional PAC-Bayes analyzes distributions directly on $\mathcal{H}$, we instead analyze distributions on $\Theta$. Thus, instead of applying the loss directly to a random hypothesis, we apply it to the output of a learning algorithm, whose inputs are a dataset and a random hyperparameter instantiation. This distinction is subtle, but important. In our framework, a random hypothesis is explicitly a function of the learning algorithm, whereas in traditional PAC-Bayes this dependence may only be implicit—for instance, if the posterior is given by random permutations of a learned hypothesis. The advantage of making the learning aspect explicit is that it isolates the source of randomness, which may help in analyzing the distribution of learned hypotheses. Indeed, it may be difficult to map the output of a randomized learning algorithm to a distribution on the hypothesis space. That said, the disadvantage of making learning explicit is that, due to the learning algorithm's dependence on the training data and hyperparameters, the generalization error could be sensitive to certain examples or hyperparameters. This condition is quantified with *algorithmic stability*, which we discuss next.

# 3 Algorithmic Stability

Informally, algorithmic stability measures the change in loss when the inputs to a learning algorithm are perturbed; a learning algorithm is stable if small perturbations lead to proportional changes in the loss. In other words, a learning algorithm should not be overly sensitive to any single input. Stability is crucial for learnability [23], and has also been linked to differentially private learning [24]. In this section, we discuss several notions of stability tailored for randomized learning algorithms. From this point on, let $D_{\mathrm{H}}(\mathbf{v}, \mathbf{v}') \triangleq \sum_{i=1}^{|\mathbf{v}|} \mathbb{1}\{v_i \neq v_i'\}$ denote the Hamming distance.

## 3.1 Definitions of Stability

The literature traditionally measures stability with respect to perturbations of the training data. We refer to this general property as *data stability*. Data stability has been defined in many ways. The following definitions, originally proposed by Elisseeff et al. [6], are designed to accommodate randomized algorithms via an expectation over the hyperparameters, $\theta \sim \mathbb{P}$.

**Definition 1** (Uniform Stability)**.** A randomized learning algorithm, $A$, is $\beta_{\mathcal{Z}}$-uniformly stable with respect to a loss function, $L$, and a distribution, $\mathbb{P}$ on $\Theta$, if

$$\sup_{S,S' \in \mathcal{Z}^n : D_{\mathrm{H}}(S,S')=1} \sup_{z \in \mathcal{Z}} \left| \mathbb{E}_{\theta \sim \mathbb{P}} \left[ L(A(S,\theta), z) - L(A(S',\theta), z) \right] \right| \leq \beta_{\mathcal{Z}}.$$

**Definition 2** (Pointwise Hypothesis Stability)**.** For a given dataset, $S$, let $S^{i,z}$ denote the result of replacing the $i^{\text{th}}$ example with example $z$. A randomized learning algorithm, $A$, is $\beta_{\mathcal{Z}}$-pointwise hypothesis stable with respect to a loss function, $L$, and a distribution, $\mathbb{P}$ on $\Theta$, if

$$\sup_{i \in \{1,\dots,n\}} \mathbb{E}_{S \sim \mathbb{D}^n} \mathbb{E}_{z \sim \mathbb{D}} \mathbb{E}_{\theta \sim \mathbb{P}} \left[ \left| L(A(S,\theta), z_i) - L(A(S^{i,z},\theta), z_i) \right| \right] \leq \beta_{\mathcal{Z}}.$$

Uniform stability measures the maximum change in loss from replacing any single training example, whereas pointwise hypothesis stability measures the expected change in loss on a random example when said example is removed from the training data. Under certain conditions, $\beta_{\mathcal{Z}}$-uniform stability implies $\beta_{\mathcal{Z}}$-pointwise hypothesis stability, but not vice versa. Thus, while uniform stability enables sharper bounds, pointwise hypothesis stability supports a wider range of learning algorithms.

In addition to data stability, we might also require stability with respect to changes in the hyperparameters. From this point forward, we will assume that the hyperparameter space, $\Theta$, decomposes into the product of $T$ subspaces, $\prod_{t=1}^{T} \Theta_t$. For instance, $\Theta$ could be the set of all sequences of example indices, $\{1,\dots,n\}^T$, such as one would sample from in SGD.

**Definition 3** (Hyperparameter Stability)**.** A randomized learning algorithm, $A$, is $\beta_{\Theta}$-uniformly stable with respect to a loss function, $L$, if

$$\sup_{S \in \mathcal{Z}^n} \sup_{z \in \mathcal{Z}} \sup_{\theta,\theta' \in \Theta : D_{\mathrm{H}}(\theta,\theta')=1} |L(A(S,\theta), z) - L(A(S,\theta'), z)| \leq \beta_{\Theta}.$$

When $A$ is both $\beta_{\mathcal{Z}}$-uniformly and $\beta_{\Theta}$-uniformly stable, we say that $A$ is $(\beta_{\mathcal{Z}}, \beta_{\Theta})$-uniformly stable.

*Remark* 1. For SGD, Definition 3 can be mapped to Bousquet and Elisseeff's [2] original definition of uniform stability using the resampled example sequence. Yet their generalization bounds would still not apply because the resampled data is not i.i.d. and SGD is not a symmetric learning algorithm.

## 3.2 Stability of Stochastic Gradient Descent

For non-vacuous generalization bounds, we will need the data stability coefficient, $\beta_{\mathcal{Z}}$, to be of order $\tilde{\mathrm{O}}(n^{-1})$. Additionally, certain results will require the hyperparameter stability coefficient, $\beta_{\Theta}$, to be of order $\tilde{\mathrm{O}}(1/\sqrt{nT})$. (If $T = \Theta(n)$, as it often is, then $\beta_{\Theta} = \tilde{\mathrm{O}}(T^{-1})$ suffices.) In this section, we review some conditions under which these requirements are satisfied by SGD. We rely on standard characterizations of the objective function—namely, convexity, Lipschitzness and smoothness—the definitions of which are deferred to Appendix A, along with all proofs from this section.

A recent study by Hardt et al. [10] proved that some special cases of SGD—when examples are sampled uniformly, with replacement—satisfy $\beta_{\mathcal{Z}}$-uniform stability (Definition 1) with $\beta_{\mathcal{Z}} = \mathrm{O}(n^{-1})$. We extend their work (specifically, [10, Theorem 3.7]) in the following result for SGD with a convex objective function, when the step size is at most inversely proportional to the current iteration.

**Proposition 1.** *Assume that the loss function, L, is $\lambda$-Lipschitz, and that the objective function, F, is convex, $\lambda$-Lipschitz and $\sigma$-smooth. Suppose SGD is run for $T$ iterations with a uniform sampling distribution, $\mathbb{P}$, and step sizes $\eta_t \in [0, \eta/t]$, for $\eta \in [0, 2/\sigma]$. Then, SGD is both $\beta_{\mathcal{Z}}$-uniformly stable and $\beta_{\mathcal{Z}}$-pointwise hypothesis stable with respect to L and $\mathbb{P}$, with*

$$\beta_{\mathcal{Z}} \leq \frac{2\lambda^2 \eta \left( \ln T + 1 \right)}{n}. \tag{1}$$

When $T = \Theta(n)$, Equation 1 is $\tilde{O}(n^{-1})$, which is acceptable for proving generalization.

If we do not assume that the objective function is convex, we can borrow a result (with small modification[2]) from Hardt et al. [10, Theorem 3.8].

**Proposition 2.** *Assume that the loss function, L, is M-bounded and $\lambda$-Lipschitz, and that the objective function, F, is $\lambda$-Lipschitz and $\sigma$-smooth. Suppose SGD is run for $T$ iterations with a uniform sampling distribution, $\mathbb{P}$, and step sizes $\eta_t \in [0, \eta/t]$, for $\eta \geq 0$. Then, SGD is both $\beta_{\mathcal{Z}}$-uniformly stable and $\beta_{\mathcal{Z}}$-pointwise hypothesis stable with respect to L and $\mathbb{P}$, with*

$$\beta_{\mathcal{Z}} \leq \left( \frac{M + (\sigma\eta)^{-1}}{n-1} \right) \left( 2\lambda^2 \eta \right)^{\frac{1}{\sigma\eta+1}} T^{\frac{\sigma\eta}{\sigma\eta+1}}. \tag{2}$$

Assuming $T = \Theta(n)$, and ignoring constants that depend on $M$, $\lambda$, $\sigma$ and $\eta$, Equation 2 reduces to $O\left( n^{-\frac{1}{\sigma\eta+1}} \right)$. As $\sigma\eta$ approaches 1, the rate becomes $O(n^{-1/2})$, which, as will become evident in Section 4, yields generalization bounds that are suboptimal, or even vacuous. However, if $\sigma\eta$ is small—say, $\eta = (10\sigma)^{-1}$—then we get $O\left( n^{-\frac{10}{11}} \right) \approx O(n^{-1})$, which suffices for generalization.

We can obtain even tighter bounds for $\beta_{\mathcal{Z}}$-pointwise hypothesis stability (Definition 2) by adopting a data-dependent view. The following result for SGD with a convex objective function is adapted from work by Kuzborskij and Lampert [13, Theorem 3].

**Proposition 3.** *Assume that the loss function, L, is $\lambda$-Lipschitz, and that the objective function, F, is convex, $\lambda$-Lipschitz and $\sigma$-smooth. Suppose SGD starts from an initial hypothesis, $h_0$, and is run for $T$ iterations with a uniform sampling distribution, $\mathbb{P}$, and step sizes $\eta_t \in [0, \eta/t]$, for $\eta \in [0, 2/\sigma]$. Then, SGD is $\beta_{\mathcal{Z}}$-pointwise hypothesis stable with respect to L and $\mathbb{P}$, with*

$$\beta_{\mathcal{Z}} \leq \frac{2\lambda\eta \left( \ln T + 1 \right) \sqrt{2\sigma \, \mathbb{E}_{z \sim \mathbb{D}}[L(h_0, z)]}}{n}. \tag{3}$$

Importantly, Equation 3 depends on the risk of the initial hypothesis, $h_0$. If $h_0$ happens to be close to a global optimum—that is, a good first guess—then Equation 3 could be tighter than Equation 1. Kuzborskij and Lampert also proved a data-dependent bound for non-convex objective functions [13, Theorem 5], which, under certain conditions, might be tighter than Equation 2. Though not presented herein, Kuzborskij and Lampert's bound is worth noting.

As we will later show, we can obtain stronger generalization guarantees by combining $\beta_{\mathcal{Z}}$-uniform stability with $\beta_{\Theta}$-uniform stability (Definition 3), provided $\beta_{\Theta} = \tilde{O}(1/\sqrt{nT})$. Prior stability analyses of SGD [10, 13] have not addressed this form of stability. Elisseeff et al. [6] proved $(\beta_{\mathcal{Z}}, \beta_{\Theta})$-uniform stability for certain bagging algorithms, but did not consider SGD. In light of Remark 1, it is tempting to map $\beta_{\Theta}$-uniform stability to Bousquet and Elisseeff's [2] uniform stability and thereby leverage their study of various regularized objective functions. However, their analysis crucially relies on exact minimization of the learning objective, whereas SGD with a finite number of steps only finds an approximate minimizer. Thus, to our knowledge, no prior work applies to this problem. As a first step, we prove uniform stability, with respect to both data and hyperparameters, for SGD with a strongly convex objective function and decaying step sizes.

**Proposition 4.** *Assume that the loss function, L, is $\lambda$-Lipschitz, and that the objective function, F, is $\gamma$-strongly convex, $\lambda$-Lipschitz and $\sigma$-smooth. Suppose SGD is run for $T$ iterations with a uniform sampling distribution, $\mathbb{P}$, and step sizes $\eta_t \triangleq (\gamma t + \sigma)^{-1}$. Then, SGD is $(\beta_{\mathcal{Z}}, \beta_{\Theta})$-uniformly stable with respect to L and $\mathbb{P}$, with*

$$\beta_{\mathcal{Z}} \leq \frac{2\lambda^2}{\gamma n} \quad and \quad \beta_{\Theta} \leq \frac{2\lambda^2}{\gamma T}. \tag{4}$$

When $T = \Theta(n)$, the $\beta_{\Theta}$ bound in Equation 4 is $O(1/\sqrt{nT})$, which supports good generalization.

# 4 Generalization Bounds

In this section, we present new generalization bounds for randomized learning algorithms. While prior work [6, 7] has addressed this topic, ours is the first PAC-Bayesian treatment (the benefits of which will be discussed momentarily). Recall that in the PAC-Bayesian framework, we fix a prior distribution, $\mathbb{P}$, on the hypothesis space, $\mathcal{H}$; then, given a sample of training data, $S \sim \mathbb{D}^n$, we learn a posterior distribution, $\mathbb{Q}$, also on $\mathcal{H}$. In our extension for randomized learning algorithms, $\mathbb{P}$ and $\mathbb{Q}$ are instead supported on the hyperparameter space, $\Theta$. Moreover, while traditional PAC-Bayes studies $\mathbb{E}_{h \sim \mathbb{Q}}[L(h, z)]$, we study the expected loss over draws of hyperparameters, $\mathbb{E}_{\theta \sim \mathbb{Q}}[L(A(S, \theta), z)]$. Our goal will be to upper-bound the generalization error of the posterior, $G(S, \mathbb{Q})$, which thereby upper-bounds the risk, $R(S, \mathbb{Q})$, by a function of the empirical risk, $\hat{R}(S, \mathbb{Q})$.

Importantly, our bounds are polynomial in $\delta^{-1}$, for a free parameter $\delta \in (0, 1)$, and hold with probability at least $1 - \delta$ over draws of a finite training dataset. This stands in contrast to related bounds [1, 10, 13, 16] that hold in expectation. While expectation bounds are useful for gaining insight into generalization behavior, high-probability bounds are sometimes preferred. Provided the loss is $M$-bounded, it is always possible to convert a high-probability bound of the form $\Pr_{S \sim \mathbb{D}^n}\{G(S, \mathbb{Q}) \le B(\delta)\} \ge 1 - \delta$ to an expectation bound of the form $\mathbb{E}_{S \sim \mathbb{D}^n}[G(S, \mathbb{Q})] \le B(\delta) + \delta M$.

Another useful property of PAC-Bayesian bounds is that they hold simultaneously for all posteriors, including those that depend on the training data. In Section 3, we assumed that hyperparameters were sampled according to a fixed distribution; for instance, sampling training example indices for SGD uniformly at random. However, in certain situations, it may be advantageous to sample according to a data-dependent distribution. Following the SGD example, suppose most training examples are *easy* to classify (e.g., far from the decision boundary), but some are *difficult* (e.g., near the decision boundary, or noisy). If we sample points uniformly at random, we might encounter mostly easy examples, which could slow progress on difficult examples. If we instead focus training on the difficult set, we might converge more quickly to an optimal hypothesis. Since our PAC-Bayesian bounds hold for all hyperparameter posteriors, we can characterize the generalization error of algorithms that optimize the posterior using the training data. Existing generalization bounds for randomized learning [6, 7], or SGD in particular [1, 10, 13, 15, 16], cannot address such algorithms. Of course, there is a penalty for overfitting the posterior to the data, which is captured by the posterior's divergence from the prior.

Our first PAC-Bayesian theorem requires the weakest stability condition, $\beta_{\mathcal{Z}}$-pointwise hypothesis stability, but the bound is sublinear in $\delta^{-1}$. Our second bound is polylogarithmic in $\delta^{-1}$, but requires the stronger stability conditions, $(\beta_{\mathcal{Z}}, \beta_{\Theta})$-uniform stability. All proofs are deferred to Appendix B.

**Theorem 1.** *Suppose a randomized learning algorithm, $A$, is $\beta_{\mathcal{Z}}$-pointwise hypothesis stable with respect to an $M$-bounded loss function, $L$, and a fixed prior, $\mathbb{P}$ on $\Theta$. Then, for any $n \ge 1$ and $\delta \in (0, 1)$, with probability at least $1 - \delta$ over draws of a dataset, $S \sim \mathbb{D}^n$, every posterior, $\mathbb{Q}$ on $\Theta$, satisfies*

$$G(S, \mathbb{Q}) \le \sqrt{\left(\frac{\chi^2(\mathbb{Q}\|\mathbb{P}) + 1}{\delta}\right)\left(\frac{2M^2}{n} + 12M\beta_{\mathcal{Z}}\right)}, \tag{5}$$

*where $\chi^2(\mathbb{Q}\|\mathbb{P}) \triangleq \mathbb{E}_{\theta \sim \mathbb{P}}\left[\left(\frac{\mathbb{Q}(\theta)}{\mathbb{P}(\theta)}\right)^2 - 1\right]$ is the $\chi^2$ divergence from $\mathbb{P}$ to $\mathbb{Q}$.*

**Theorem 2.** *Suppose a randomized learning algorithm, $A$, is $(\beta_{\mathcal{Z}}, \beta_{\Theta})$-uniformly stable with respect to an $M$-bounded loss function, $L$, and a fixed product measure, $\mathbb{P}$ on $\Theta = \prod_{t=1}^{T} \Theta_t$. Then, for any $n \ge 1$, $T \ge 1$ and $\delta \in (0, 1)$, with probability at least $1 - \delta$ over draws of a dataset, $S \sim \mathbb{D}^n$, every posterior, $\mathbb{Q}$ on $\Theta$, satisfies*

$$G(S, \mathbb{Q}) \le \beta_{\mathcal{Z}} + \sqrt{2\left(D_{\text{KL}}(\mathbb{Q}\|\mathbb{P}) + \ln\frac{2}{\delta}\right)\left(\frac{(M + 2n\beta_{\mathcal{Z}})^2}{n} + 4T\beta_{\Theta}^2\right)}, \tag{6}$$

*where $D_{\text{KL}}(\mathbb{Q}\|\mathbb{P}) \triangleq \mathbb{E}_{\theta \sim \mathbb{Q}}\left[\ln\left(\frac{\mathbb{Q}(\theta)}{\mathbb{P}(\theta)}\right)\right]$ is the KL divergence from $\mathbb{P}$ to $\mathbb{Q}$.*

Since Theorems 1 and 2 hold simultaneously for all hyperparameter posteriors, they provide generalization guarantees for SGD with any sampling distribution. Note that the stability requirements only need to be satisfied by a fixed product measure, such as a uniform distribution. This simple

sampling distribution can have $\left(\mathrm{O}(n^{-1}), \mathrm{O}(T^{-1})\right)$-uniform stability under certain conditions, as demonstrated in Section 3.2. In the following, we apply Theorem 2 to SGD with a strongly convex objective function, leveraging Proposition 4 to upper-bound the stability coefficients.

**Corollary 1.** *Assume that the loss function, $L$, is $M$-bounded and $\lambda$-Lipschitz, and that the objective function, $F$, is $\gamma$-strongly convex, $\lambda$-Lipschitz and $\sigma$-smooth. Let $\mathbb{P}$ denote a uniform prior on $\{1, \ldots, n\}^T$. Then, for any $n \geq 1$, $T \geq 1$ and $\delta \in (0,1)$, with probability at least $1 - \delta$ over draws of a dataset, $S \sim \mathbb{D}^n$, SGD with step sizes $\eta_t \triangleq (\gamma t + \sigma)^{-1}$ and any posterior sampling distribution, $\mathbb{Q}$ on $\{1, \ldots, n\}^T$, satisfies*

$$G(S, \mathbb{Q}) \leq \frac{2\lambda^2}{\gamma n} + \sqrt{2\left(D_{\mathrm{KL}}(\mathbb{Q}\|\mathbb{P}) + \ln\frac{2}{\delta}\right)\left(\frac{(M + 4\lambda^2/\gamma)^2}{n} + \frac{16\lambda^4}{\gamma^2 T}\right)}.$$

When the divergence is polylogarithmic in $n$, and $T = \Theta(n)$, the generalization bound is $\tilde{\mathrm{O}}(n^{-1/2})$. In the special case of uniform sampling, the KL divergence is zero, yielding a $\mathrm{O}(n^{-1/2})$ bound.

Importantly, Theorem 1 does not require hyperparameter stability, and is therefore of interest for analyzing non-convex objective functions, since it is not known whether uniform hyperparameter stability can be satisfied without (strong) convexity. One can use Equation 2 (or [13, Theorem 5]) to upper-bound $\beta_{\mathcal{Z}}$ in Equation 5 and thereby obtain a generalization bound for SGD with a non-convex objective function, such as neural network training. We leave this substitution to the reader.

Equation 6 holds with high probability over draws of a dataset, but the generalization error is an expected value over draws of hyperparameters. To obtain a bound that holds with high probability over draws of both data and hyperparameters, we consider posteriors that are product measures.

**Theorem 3.** *Suppose a randomized learning algorithm, $A$, is $(\beta_{\mathcal{Z}}, \beta_{\Theta})$-uniformly stable with respect to an $M$-bounded loss function, $L$, and a fixed product measure, $\mathbb{P}$ on $\Theta = \prod_{t=1}^{T} \Theta_t$. Then, for any $n \geq 1$, $T \geq 1$ and $\delta \in (0,1)$, with probability at least $1 - \delta$ over draws of a dataset, $S \sim \mathbb{D}^n$, and hyperparameters, $\theta \sim \mathbb{Q}$, from any posterior product measure, $\mathbb{Q}$ on $\Theta$,*

$$G(S, \theta) \leq \beta_{\mathcal{Z}} + \beta_{\Theta}\sqrt{2\,T\ln\frac{2}{\delta}} + \sqrt{2\left(D_{\mathrm{KL}}(\mathbb{Q}\|\mathbb{P}) + \ln\frac{4}{\delta}\right)\left(\frac{(M + 2n\beta_{\mathcal{Z}})^2}{n} + 4T\beta_{\Theta}^2\right)}. \quad (7)$$

If $\beta_{\Theta} = \tilde{\mathrm{O}}(1/\sqrt{nT})$, then $\beta_{\Theta}\sqrt{2\,T\ln\frac{2}{\delta}}$ vanishes at a rate of $\tilde{\mathrm{O}}(n^{-1/2})$. We can apply Theorem 3 to SGD in the same way we applied Theorem 2 in Corollary 1. Further, note that a uniform distribution is a product distribution. Thus, if we eschew optimizing the posterior, then the KL divergence disappears, leaving a $\mathrm{O}(n^{-1/2})$ derandomized generalization bound for SGD with uniform sampling.[3]

# 5    Adaptive Sampling for Stochastic Gradient Descent

The PAC-Bayesian theorems in Section 4 motivate data-dependent posterior distributions on the hyperparameter space. Intuitively, certain posteriors may improve, or speed up, learning from a given dataset. For instance, suppose certain training examples are considered valuable for reducing empirical risk; then, a sampling posterior for SGD should weight those examples more heavily than others, so that the learning algorithm can, probabilistically, focus its attention on the valuable examples. However, a posterior should also try to stay close to the prior, to control the divergence penalty in the generalization bounds.

Based on this idea, we propose a sampling procedure for SGD (or any variant thereof) that constructs a posterior based on the training data, balancing the utility of the sampling distribution with its divergence from a uniform prior. The algorithm operates alongside the learning algorithm, iteratively generating the posterior as a sequence of conditional distributions on the training data. Each iteration of training generates a new distribution conditioned on the previous iterations, so the posterior dynamically adapts to training. We therefore call our algorithm *adaptive sampling SGD*.

**Algorithm 1** Adaptive Sampling SGD

---

**Require:** Examples, $(z_1, \ldots, z_n) \in \mathcal{Z}^n$; initial hypothesis, $h_0 \in \mathcal{H}$; update rule, $U_t : \mathcal{H} \times \mathcal{Z} \to \mathcal{H}$; utility function, $f : \mathcal{Z} \times \mathcal{H} \to \mathbb{R}$; amplitude, $\alpha \geq 0$; decay, $\tau \in (0, 1)$.

1: $(q_1, \ldots, q_n) \leftarrow \mathbf{1}$        ▷ Initialize sampling weights uniformly
2: **for** $t = 1, \ldots, T$ **do**
3:     $i_t \sim \mathbb{Q}_t \propto (q_1, \ldots, q_n)$        ▷ Draw index $i_t$ proportional to sampling weights
4:     $h_t \leftarrow U_t(h_{t-1}, z_{i_t})$        ▷ Update hypothesis
5:     $q_{i_t} \leftarrow q_{i_t}^\tau \exp\left(\alpha f(z_{i_t}, h_t)\right)$        ▷ Update sampling weight for $i_t$

6: **return** $h_T$

---

Algorithm 1 maintains a set of nonnegative sampling weights, $(q_1, \ldots, q_n)$, which define a distribution on the dataset. The posterior probability of the $i^{\text{th}}$ example in the $t^{\text{th}}$ iteration, given the previous iterations, is proportional to the $i^{\text{th}}$ weight: $\mathbb{Q}_t(i) \triangleq \mathbb{Q}(i_t = i \mid i_1, \ldots, i_{t-1}) \propto q_i$. The sampling weights are initialized to 1, thereby inducing a uniform distribution. At each iteration, we draw an index, $i_t \sim \mathbb{Q}_t$, and use example $z_{i_t}$ to update the hypothesis. We then update the weight for $i_t$ multiplicatively as $q_{i_t} \leftarrow q_{i_t}^\tau \exp\left(\alpha f(z_{i_t}, h_t)\right)$, where: $f(z_{i_t}, h_t)$ is a *utility* function of the chosen example and current hypothesis; $\alpha \geq 0$ is an *amplitude* parameter, which controls the aggressiveness of the update; and $\tau \in (0, 1)$ is a *decay* parameter, which lets $q_i$ gradually forget past updates.

The multiplicative weight update (line 5) can be derived by choosing a sampling distribution for the next iteration, $t + 1$, that maximizes the expected utility while staying close to a reference distribution. Consider the following constrained optimization problem:

$$\max_{\mathbb{Q}_{t+1} \in \Delta^n} \sum_{i=1}^n \mathbb{Q}_{t+1}(i) f(z_i, h_t) - \frac{1}{\alpha} D_{\text{KL}}(\mathbb{Q}_{t+1} \| \mathbb{Q}_t^\tau). \tag{8}$$

The term $\sum_{i=1}^n \mathbb{Q}_{t+1}(i) f(z_i, h_t)$ is the expected utility under the new distribution, $\mathbb{Q}_{t+1}$. This is offset by the KL divergence, which acts as a regularizer, penalizing $\mathbb{Q}_{t+1}$ for diverging from a reference distribution, $\mathbb{Q}_t^\tau$, where $\mathbb{Q}_t^\tau(i) \propto q_i^\tau$. The decay parameter, $\tau$, controls the *temperature* of the reference distribution, allowing it to interpolate between the current distribution ($\tau = 1$) and a uniform distribution ($\tau = 0$). The amplitude parameter, $\alpha$, scales the influence of the regularizer relative to the expected utility. We can solve Equation 8 analytically using the method of Lagrange multipliers, which yields

$$\mathbb{Q}_{t+1}^\star(i) \propto \mathbb{Q}_t^\tau(i) \exp\left(\alpha f(z_{i_t}, h_t) - 1\right) \propto q_i^\tau \exp\left(\alpha f(z_{i_t}, h_t)\right).$$

Updating $q_i$ for all $i = 1, \ldots, n$ is impractical for large $n$, so we approximate the above solution by only updating the weight for the last sampled index, $i_t$, effectively performing coordinate ascent.

The idea of tuning the empirical data distribution through multiplicative weight updates is reminiscent of AdaBoost [8] and focused online learning [22], but note that Algorithm 1 learns a single hypothesis, not an ensemble. In this respect, it is similar to SelfieBoost [21]. One could also draw parallels to exponentiated gradient dual coordinate ascent [4]. Finally, note that when the gradient estimate is unbiased (i.e., weighted by the inverse sampling probability), we obtain a variant of importance sampling SGD [25], though we do not necessarily need unbiased gradient estimates.

It is important to note that we do not actually need to compute the full posterior distribution—which would take $\text{O}(n)$ time per iteration—in order to sample from it. Indeed, using an algorithm and data structure described in Appendix C, we can sample from and update the distribution in $\text{O}(\log n)$ time, using $\text{O}(n)$ space. Thus, the additional iteration complexity of adaptive sampling is logarithmic in the size of the dataset, which suitably efficient for learning from large datasets.

In practice, SGD is typically applied with *mini-batching*, whereby multiple examples are drawn at each iteration, instead of just one. Given the massive parallelism of today's computing hardware, mini-batching is simply a more efficient way to process a dataset, and can result in more accurate gradient estimates than single-example updates. Though Algorithm 1 is stated for single-example updates, it can be modified for mini-batching by replacing line 3 with multiple independent draws from $\mathbb{Q}_t$, and line 5 with sampling weight updates for each unique[4] example in the mini-batch.

## 5.1 Divergence Analysis

Recall that our generalization bounds use the posterior's divergence from a fixed prior to penalize the posterior for overfitting the training data. Thus, to connect Algorithm 1 to our bounds, we analyze the adaptive posterior's divergence from a uniform prior on $\{1, \ldots, n\}^T$. This quantity reflects the potential cost, in generalization performance, of adaptive sampling. The goal of this section is to upper-bound the KL divergence resulting from Algorithm 1 in terms of interpretable, data-dependent quantities. All proofs are deferred to Appendix D.

Our analysis requires introducing some notation. Given a sequence of sampled indices, $(i_1, \ldots, i_t)$, let $N_{i,t} \triangleq |\{t' : t' < t, i_{t'} = i\}|$ denote the number of times that index $i$ was chosen *before* iteration $t$. Let $O_{i,j}$ denote the $j^{\text{th}}$ iteration in which $i$ was chosen; for instance, if $i$ was chosen at iterations 13 and 47, then $O_{i,1} = 13$ and $O_{i,2} = 47$. With these definitions, we can state the following bound, which exposes the influences of the utility function, amplitude and decay on the KL divergence.

**Theorem 4.** *Fix a uniform prior, $\mathbb{P}$, a utility function, $f : \mathcal{Z} \times \mathcal{H} \to \mathbb{R}$, an amplitude, $\alpha \geq 0$, and a decay, $\tau \in (0, 1)$. If Algorithm 1 is run for $T$ iterations, then its posterior, $\mathbb{Q}$, satisfies*

$$D_{\text{KL}}(\mathbb{Q}\|\mathbb{P}) \leq \sum_{t=2}^{T} \mathop{\mathbb{E}}_{(i_1,\ldots,i_t)\sim\mathbb{Q}} \frac{\alpha}{n} \sum_{i=1}^{n} \left[ \sum_{j=1}^{N_{i_t,t}} f(z_{i_t}, h_{O_{i_t,j}}) \tau^{N_{i_t,t}-j} - \sum_{k=1}^{N_{i,t}} f(z_i, h_{O_{i,k}}) \tau^{N_{i,t}-k} \right]. \quad (9)$$

Equation 9 can be interpreted as measuring, on average, how the cumulative past utilities of each sampled index, $i_t$, differ from the cumulative utilities of any other index, $i$.[5] When the posterior becomes too focused on certain examples, this difference is large. The accumulated utilities decay exponentially, with the rate of decay controlled by $\tau$. The amplitude, $\alpha$, scales the entire bound, which means that aggressive posterior updates may adversely affect generalization.

An interesting special case of Theorem 4 is when the utility function is nonnegative, which results in a simpler, more interpretable bound.

**Theorem 5.** *Fix a uniform prior, $\mathbb{P}$, a nonnegative utility function, $f : \mathcal{Z} \times \mathcal{H} \to \mathbb{R}_+$, an amplitude, $\alpha \geq 0$, and a decay, $\tau \in (0, 1)$. If Algorithm 1 is run for $T$ iterations, then its posterior, $\mathbb{Q}$, satisfies*

$$D_{\text{KL}}(\mathbb{Q}\|\mathbb{P}) \leq \frac{\alpha}{1-\tau} \sum_{t=1}^{T-1} \mathop{\mathbb{E}}_{(i_1,\ldots,i_t)\sim\mathbb{Q}} \left[ f(z_{i_t}, h_t) \right]. \quad (10)$$

Equation 10 is simply the sum of expected utilities computed over $T-1$ iterations of training, scaled by $\alpha/(1-\tau)$. The implications of this bound are interesting when the utility function is defined as the loss, $f(z, h) \triangleq L(h, z)$; then, if SGD quickly converges to a hypothesis with low *maximal* loss on the training data, it can reduce the generalization error.[6] The caveat is that tuning the amplitude or decay to speed up convergence may actually counteract this effect.

It is worth noting that similar guarantees hold for a mini-batch variant of Algorithm 1. The bounds are essentially unchanged, modulo notational intricacies.

## 6 Experiments

To demonstrate the effectiveness of Algorithm 1, we conducted several experiments with the CIFAR-10 dataset [12]. This benchmark dataset contains 60,000 $(32\times32)$-pixel RGB images from 10 object classes, with a standard, static partitioning into 50,000 training examples and 10,000 test examples.

We specified the hypothesis class as the following convolutional neural network architecture: 32 $(3 \times 3)$ filters with rectified linear unit (ReLU) activations in the first and second layers, followed by $(2 \times 2)$ max-pooling and 0.25 dropout[7]; 64 $(3 \times 3)$ filters with ReLU activations in the third and fourth layers, again followed by $(2 \times 2)$ max-pooling and 0.25 dropout; finally, a fully-connected, 512-unit layer with ReLU activations and 0.5 dropout, followed by a fully-connected, 10-output softmax layer. We trained the network using the cross-entropy loss. We emphasize that our goal was

not to achieve state-of-the-art results on the dataset; rather, to evaluate Algorithm 1 in a simple, yet realistic, application.

Following the intuition that sampling should focus on difficult examples, we experimented with two utility functions for Algorithm 1 based on common loss functions. For an example $z = (x, y)$, with $h(x, y)$ denoting the predicted probability of label $y$ given input $x$ under hypothesis $h$, let

$$f_0(z, h) \triangleq \mathbb{1}\{\arg\max_{y' \in \mathcal{Y}} h(x, y') \neq y\} \quad \text{and} \quad f_1(z, h) \triangleq 1 - h(x, y).$$

The first utility function, $f_0$, is the 0-1 loss; the second, $f_1$, is the $L_1$ loss, which accounts for uncertainty in the most likely label. We combined these utility functions with two parameter update rules: standard SGD with decreasing step sizes, $\eta_t \triangleq \eta/(1+\nu t) \leq \eta/(\nu t)$, for $\eta > 0$ and $\nu > 0$; and AdaGrad [5], a variant of SGD that automatically tunes a separate step size for each parameter. We used mini-batches of 100 examples per update. The combination of utility functions and update rules yields four adaptive sampling algorithms: AdaSamp-01-SGD, AdaSamp-01-AdaGrad, AdaSamp-L1-SGD and AdaSamp-L1-AdaGrad. We compared these to their uniform sampling counterparts, Unif-SGD and Unif-AdaGrad.

We tuned all hyperparameters using random subsets of the training data for cross-validation. We then ran 10 trials of training and testing, using different seeds for the pseudorandom number generator at each trial to generate different random initializations[8] and training sequences. Figures 1a and 1b plot learning curves of the average cross-entropy and accuracy, respectively, on the training data; Figure 1c plots the average accuracy on the test data. We found that all adaptive sampling variants reduced empirical risk (increased training accuracy) faster than their uniform sampling counterparts. Further, AdaGrad with adaptive sampling exhibited modest, yet consistent, improvements in test accuracy in early iterations of training. Figure 1d illustrates the effect of varying the amplitude parameter, $\alpha$. Higher values of $\alpha$ led to faster empirical risk reduction, but lower test accuracy—a sign of overfitting the posterior to the data, which concurs with Theorems 4 and 5 regarding the influence of $\alpha$ on the KL divergence. Figure 1e plots the KL divergence from the conditional prior, $\mathbb{P}_t$, to the conditional posterior, $\mathbb{Q}_t$, given sampled indices $(i_1, \ldots, i_{t-1})$; i.e., $D_{\mathrm{KL}}(\mathbb{Q}_t \| \mathbb{P}_t)$. The sampling distribution quickly diverged in early iterations, to focus on examples where the model erred, then gradually converged to a uniform distribution as the empirical risk converged.

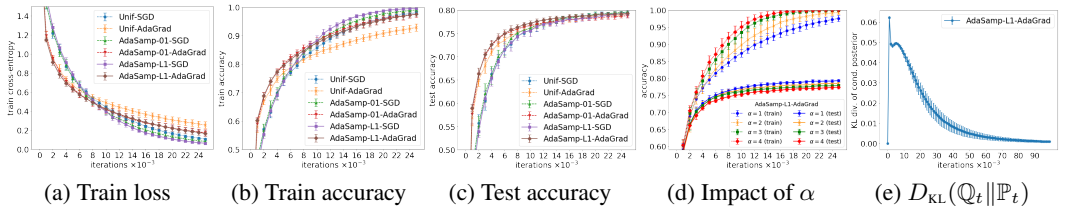

| (a) Train loss | (b) Train accuracy | (c) Test accuracy | (d) Impact of $\alpha$ | (e) $D_{\mathrm{KL}}(\mathbb{Q}_t \| \mathbb{P}_t)$ |

Figure 1: Experimental results on CIFAR-10, averaged over 10 random initializations and training runs. (Best viewed in color.) Figure 1a plots learning curves of training cross-entropy (lower is better). Figures 1b and 1c, respectively, plot train and test accuracies (higher is better). Figure 1d highlights the impact of the amplitude parameter, $\alpha$, on accuracy. Figure 1e plots the KL divergence from the conditional prior, $\mathbb{P}_t$, to the conditional posterior, $\mathbb{Q}_t$, given sampled indices $(i_1, \ldots, i_{t-1})$.

# 7 Conclusions and Future Work

We presented new generalization bounds for randomized learning algorithms, using a novel combination of PAC-Bayes and algorithmic stability. The bounds inspired an adaptive sampling algorithm for SGD that dynamically updates the sampling distribution based on the training data and model. Experimental results with this algorithm indicate that it can reduce empirical risk faster than uniform sampling while also improving out-of-sample accuracy. Future research could investigate different utility functions and distribution updates, or explore the connections to related algorithms. We are also interested in providing stronger generalization guarantees, with polylogarithmic dependence on $\delta^{-1}$, for non-convex objective functions, but proving $\tilde{O}(1/\sqrt{nT})$-uniform hyperparameter stability without (strong) convexity is difficult. We hope to address this problem in future work.

## Footnotes

[1] Accommodating unbounded loss functions is possible [11], but requires additional assumptions.

[2]Hardt et al.'s definition of stability and theorem statement differ slightly from ours. See Appendix A.1.

[3]We can achieve the same result by pairing Proposition 4 with Elisseeff et al.'s generalization bound for algorithms with $(\beta_{\mathcal{Z}}, \beta_{\Theta})$-uniform stability [6, Theorem 15]. However, Elisseeff et al.'s bound only applies to fixed product measures on $\Theta$, whereas Theorem 3 applies more generally to any posterior product measure, and when $\mathbb{P} = \mathbb{Q}$, Equation 7 is within a constant factor of Elisseeff et al.'s bound.

[4] If an example is drawn multiple times in a mini-batch, its sampling weight is only updated once.

[5]When $N_{i,t} = 0$ (i.e., $i$ has not yet been sampled), a summation over $j = 1, \ldots, N_{i,t}$ evaluates to zero.

[6]This interpretation concurs with ideas in [10, 22].

[7]It can be shown that dropout improves data stability [10, Lemma 4.4].

[8]Each training algorithm started from the same initial hypothesis.

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
