[Supplementary Material · london-nips17-supp.pdf]

# A PAC-Bayesian Analysis of Randomized Learning with Application to Stochastic Gradient Descent
## *** Supplemental Material ***

**Ben London**
blondon@amazon.com
Amazon AI

## A    Proofs from Section 3

The stability bounds in Section 3 require several characterizations of a loss or objective function. In the following definitions, we consider generic functions of the form $\varphi : \mathcal{H} \times \mathcal{Z} \to \mathbb{R}$. Since we are only interested in how a function behaves with respect to $\mathcal{H}$, we specify the definitions accordingly.

**Definition 4** (Convexity). A differentiable function, $\varphi : \mathcal{H} \times \mathcal{Z} \to \mathbb{R}$, is convex (in $\mathcal{H}$) if

$$\forall h, h' \in \mathcal{H}, \; \forall z \in \mathcal{Z}, \; \langle \nabla \varphi(h, z), h' - h \rangle \leq \varphi(h', z) - \varphi(h, z).$$

Further, $\varphi$ is $\gamma$-strongly convex (with respect to the 2-norm) if

$$\frac{\gamma}{2} \left\| h' - h \right\|^2 + \langle \nabla \varphi(h, z), h' - h \rangle \leq \varphi(h', z) - \varphi(h, z).$$

**Definition 5** (Lipschitzness). A function, $\varphi : \mathcal{H} \times \mathcal{Z} \to \mathbb{R}$, is $\lambda$-Lipschitz (in $\mathcal{H}$) if

$$\sup_{h, h' \in \mathcal{H}} \sup_{z \in \mathcal{Z}} \frac{|\varphi(h, z) - \varphi(h', z)|}{\|h - h'\|} \leq \lambda. \tag{11}$$

If $\varphi$ is differentiable, then Equation 11 is equivalent to

$$\sup_{h \in \mathcal{H}} \sup_{z \in \mathcal{Z}} \|\nabla \varphi(h, z)\| \leq \lambda.$$

**Definition 6** (Smoothness). A differentiable function, $\varphi : \mathcal{H} \times \mathcal{Z} \to \mathbb{R}$, is $\sigma$-smooth (in $\mathcal{H}$) if

$$\sup_{h, h' \in \mathcal{H}} \sup_{z \in \mathcal{Z}} \frac{\|\nabla \varphi(h, z) - \nabla \varphi(h', z)\|}{\|h - h'\|} \leq \sigma.$$

Smoothness is a form of Lipschitzness; a function is $\sigma$-smooth if its gradient is $\sigma$-Lipschitz.

### A.1    Proof of Propositions 1 to 3

Propositions 1 to 3 extend work by Hardt et al. [5] and Kuzborskij and Lampert [8], whose definitions of data stability differ slightly from ours (which are taken from [3]). To reconcile our definition of $\beta_{\mathcal{Z}}$-uniform stability with Hardt et al.'s, which does not involve an absolute value, observe that

$$\sup_{S, S', z} \left| \mathop{\mathbb{E}}_{\theta \sim \mathbb{P}} \left[ L(A(S, \theta), z) - L(A(S', \theta), z) \right] \right| = \sup_{S, S', z} \mathop{\mathbb{E}}_{\theta \sim \mathbb{P}} \left[ L(A(S, \theta), z) - L(A(S', \theta), z) \right]$$

by the symmetry of the supremum over $S$ and $S'$. Kuzborskij and Lampert's definition of hypothesis stability equates to our *pointwise* hypothesis stability, though they do not include an absolute value inside the expectation over $\theta \sim \mathbb{P}$. Nonetheless, since the loss function is always assumed to be

$\lambda$-Lipschitz, this distinction does not matter. Indeed, all existing stability proofs for SGD implicitly leverage the following upper bound:

$$\underset{\theta\sim\mathbb{P}}{\mathbb{E}}[L(A(S,\theta),z) - L(A(S',\theta),z)] \leq \underset{\theta\sim\mathbb{P}}{\mathbb{E}}[|L(A(S,\theta),z) - L(A(S',\theta),z)|]$$

$$\leq \lambda \underset{\theta\sim\mathbb{P}}{\mathbb{E}}[\|A(S,\theta) - A(S',\theta)\|]. \tag{12}$$

Equation 12 implies that Kuzborskij and Lampert's proofs hold for our definition of pointwise hypothesis stability; we simply start the proof from the right-hand side of the first inequality. By the same logic, we can convert existing proofs of $\beta_{\mathcal{Z}}$-uniform stability to proofs of $\beta_{\mathcal{Z}}$-pointwise hypothesis stability. Moreover, Equation 12 lets us distinguish between the loss function, $L$, and the objective function, $F$, which is optimized by $A$; though [5, 8] do not make this distinction, their results hold when $L \neq F$ because they assume Lipschitzness.

Using the above reasoning, we therefore arrive at the following adaptations, which will be used to prove Propositions 1 and 3.

**Lemma 1** (adapted from [5, Theorem 3.7]). *Assume that the loss function, $L$, is $\lambda$-Lipschitz, and that the objective function, $F$, is convex, $\lambda$-Lipschitz and $\sigma$-smooth. Suppose SGD is run for $T$ iterations with a uniform sampling distribution, $\mathbb{P}$, and step sizes $\eta_t \in [0, 2/\sigma]$. Then, SGD is both $\beta_{\mathcal{Z}}$-uniformly stable and $\beta_{\mathcal{Z}}$-pointwise hypothesis stable with respect to $L$ and $\mathbb{P}$, with*

$$\beta_{\mathcal{Z}} \leq \frac{2\lambda^2}{n} \sum_{t=1}^{T} \eta_t.$$

**Lemma 2** (adapted from [8, Theorem 3]). *Assume that the loss function, $L$, is $\lambda$-Lipschitz, and that the objective function, $F$, is convex, $\lambda$-Lipschitz and $\sigma$-smooth. Suppose SGD starts from an initial hypothesis, $h_0$, and is run for $T$ iterations with a uniform sampling distribution, $\mathbb{P}$, and step sizes $\eta_t \in [0, 2/\sigma]$. Then, SGD is $\beta_{\mathcal{Z}}$-pointwise hypothesis stable with respect to $L$ and $\mathbb{P}$, with*

$$\beta_{\mathcal{Z}} \leq \frac{2\lambda\sqrt{2\sigma\,\mathbb{E}_{z\sim\mathbb{D}}[L(h_0,z)]}}{n} \sum_{t=1}^{T} \eta_t.$$

If $\eta \leq 2/\sigma$, then $\eta_t \leq \eta/t \leq 2/\sigma$ for all $t \geq 1$. We thus have from Lemma 1 that

$$\beta_{\mathcal{Z}} \leq \frac{2\lambda^2\eta}{n} \sum_{t=1}^{T} \frac{1}{t} \leq \frac{2\lambda^2\eta}{n}(\ln T + 1),$$

which proves Proposition 1. The last inequality follows from the fact that the $T^{\text{th}}$ harmonic number, $\sum_{t=1}^{T} \frac{1}{t}$, is upper-bounded by $\ln T + 1$. We obtain Proposition 3 from Lemma 2 using an identical proof.

Proposition 2 follows from [5, Theorem 3.8] with a few small modifications. As previously mentioned, we can use Equation 12 to reconcile definitional differences, distinguish $L$ from $F$, and adapt the proof for pointwise hypothesis complexity. We also assume that $L$ is $M$-bounded instead of 1-bounded, so we use $\sup_{h,z} L(h,z) \leq M$ in the proof (see [4, Lemma 3.11]).

## A.2 Proof of Proposition 4

We characterize SGD updates using the following definition, borrowed from Hardt et al. [5].

**Definition 7** (Expansivity). An update rule, $U : \mathcal{H} \times \mathcal{Z} \to \mathcal{H}$, is $\alpha$-expansive if

$$\sup_{h,h'\in\mathcal{H}} \sup_{z\in\mathcal{Z}} \frac{\|U(h,z) - U(h',z)\|}{\|h - h'\|} \leq \alpha.$$

We say that $U$ is contractive if $\alpha \leq 1$.

Expansivity is yet another form of Lipschitzness; an update rule is $\alpha$-expansive if it is $\alpha$-Lipschitz.

We begin our proof with a fundamental technical lemma.

**Lemma 3.** *Assume that the objective function, $F$, is $\lambda$-Lipschitz. Further, assume that each SGD update, $U_t$, is $\alpha_t$-expansive. If SGD is run for $T$ iterations on two sequences of examples that differ at a single iteration, $k$, then the resulting learned hypotheses, $h_T$ and $h'_T$, satisfy*

$$\|h_T - h'_T\| \leq 2\lambda\eta_k \prod_{t=k+1}^{T} \alpha_t.$$

*Proof.* For the first $k-1$ iterations of SGD, the example sequences are the same; therefore, so are the learned weights. On processing the $k^{\text{th}}$ example, the weights may diverge, but we will show that the divergence is bounded, due to the Lipschitz property. For every iteration after $k$, the weights may continue to follow different trajectories, but the expansivity property lets us bound the difference of the final, learned weights.

Starting at $T$ and recursing backward, we have that

$$\|h_T - h'_T\| \leq \|h_{T-1} - h'_{T-1}\| \alpha_T \leq \ldots \leq \|h_k - h'_k\| \prod_{t=k+1}^{T} \alpha_t. \tag{13}$$

Then, expanding the $k^{\text{th}}$ update,

$$
\begin{aligned}
\|h_k - h'_k\| &= \|h_{k-1} - \eta_k \nabla F(h_{k-1}, z_k) - h_{k-1} + \eta_k \nabla F(h_{k-1}, z'_k)\| \\
&\leq \|\eta_k \nabla F(h_{k-1}, z_k)\| + \|\eta_k \nabla F(h_{k-1}, z'_k)\| \\
&\leq 2\eta_k \lambda. 
\end{aligned} \tag{14}
$$

Combining Equations 13 and 14 completes the proof. $\square$

We can now prove Proposition 4. First, note that $\eta_t \leq 1/\sigma$ for all $t \geq 1$. As noted by Hardt et al. [4, proof of Theorem 3.9], due to the strong convexity of the objective function, this step size guarantees that each update is contractive with coefficient $1 - \eta_t\gamma = 1 - (t + \sigma/\gamma)^{-1}$. Moreover [4, proof of Theorem 3.10],

$$
\begin{aligned}
\mathbb{E}_{\theta \sim \mathbb{P}}[\|h_T - h'_T\|] &\leq \sum_{k=1}^{T} \left( \prod_{t=k+1}^{T} (1 - \eta_t\gamma) \right) \eta_k \cdot \frac{2\lambda}{n} \\
&= \sum_{k=1}^{T} \left( \prod_{t=k+1}^{T} \left( 1 - \frac{1}{t + \sigma/\gamma} \right) \right) \frac{1}{k + \sigma/\gamma} \cdot \frac{2\lambda}{\gamma n} \\
&= \sum_{k=1}^{T} \frac{k + \sigma/\gamma}{T + \sigma/\gamma} \cdot \frac{1}{k + \sigma/\gamma} \cdot \frac{2\lambda}{\gamma n} \\
&= \frac{T}{T + \sigma/\gamma} \cdot \frac{2\lambda}{\gamma n} \leq \frac{2\lambda}{\gamma n}.
\end{aligned} \tag{15}
$$

Combining Equations 12 and 15 yields an upper bound on the data stability coefficient, $\beta_{\mathcal{Z}} \leq \frac{2\lambda^2}{\gamma n}$.

Now, suppose the example sequence is perturbed at any index $k$. Via Lemma 3, we have that

$$
\begin{aligned}
\|h_T - h'_T\| &\leq 2\lambda\eta_k \prod_{t=k+1}^{T} (1 - \eta_t\gamma) \\
&= \frac{2\lambda}{\gamma} \cdot \frac{1}{k + \sigma/\gamma} \prod_{t=k+1}^{T} \left( 1 - \frac{1}{t + \sigma/\gamma} \right) \\
&= \frac{2\lambda}{\gamma} \cdot \frac{1}{k + \sigma/\gamma} \cdot \frac{k + \sigma/\gamma}{T + \sigma/\gamma} \leq \frac{2\lambda}{\gamma T},
\end{aligned}
$$

which we combine with the Lipschitz property (Equation 11) to obtain $\beta_\Theta \leq \frac{2\lambda^2}{\gamma T}$.

# B  Proofs from Section 4

## B.1  Stability of the Generalization Error

Our analysis in Section 4 uses stability to bound the moments and moment-generating function of the generalization error. To enable these proofs, we first derive some technical lemmas that relate stability in the loss to the stability in the generalization error. The first lemma applies to data stability; the second, to hyperparameter stability.

**Lemma 4.** *If $A$ is $\beta_{\mathcal{Z}}$-uniformly stable with respect to an $M$-bounded loss function, $L$, and a distribution, $\mathbb{P}$, then, for any $S, S' \in \mathcal{Z}^n : D_{\mathrm{H}}(S, S') = 1$,*

$$G(S, \mathbb{P}) - G(S', \mathbb{P}) \leq 2\beta_{\mathcal{Z}} + \frac{M}{n}.$$

*Proof.* Observe that the difference of generalization errors decomposes as

$$G(S, \mathbb{P}) - G(S', \mathbb{P}) = (R(S, \mathbb{P}) - R(S', \mathbb{P})) + (\hat{R}(S', \mathbb{P}) - \hat{R}(S, \mathbb{P})). \tag{16}$$

We will upper-bound each difference separately. First, using linearity of expectation and $\beta_{\mathcal{Z}}$-uniform stability, we have that

$$R(S, \mathbb{P}) - R(S', \mathbb{P}) = \mathop{\mathbb{E}}_{z \sim \mathbb{D}} \mathop{\mathbb{E}}_{\theta \sim \mathbb{P}} [L(A(S, \theta), z) - L(A(S', \theta), z)] \leq \beta_{\mathcal{Z}}. \tag{17}$$

Then, without loss of generality, assume that $S'$ differs from $S$ at the $i^{\mathrm{th}}$ example, denoted $z_i'$. Using $\beta_{\mathcal{Z}}$-uniform stability again,

$$
\begin{aligned}
\hat{R}(S', \mathbb{P}) - \hat{R}(S, \mathbb{P}) &= \frac{1}{n} \sum_{j \neq i} \mathop{\mathbb{E}}_{\theta \sim \mathbb{P}} [L(A(S', \theta), z_j) - L(A(S, \theta), z_j)] \\
&\quad + \frac{1}{n} \mathop{\mathbb{E}}_{\theta \sim \mathbb{P}} [L(A(S', \theta), z_i') - L(A(S, \theta), z_i)] \\
&\leq \frac{1}{n} \sum_{j \neq i} \beta_{\mathcal{Z}} + \frac{M}{n} \leq \beta_{\mathcal{Z}} + \frac{M}{n}.
\end{aligned} \tag{18}
$$

Combining Equations 16 to 18 completes the proof.  $\square$

**Lemma 5.** *If $A$ is $\beta_{\Theta}$-uniformly stable with respect to a loss function, $L$, then, for any $S \in \mathcal{Z}^n$ and $\theta, \theta' \in \Theta : D_{\mathrm{H}}(\theta, \theta') = 1$,*

$$G(S, \theta) - G(S, \theta') \leq 2\beta_{\Theta}.$$

*Proof.* The proof is almost identical to that of Lemma 4. First, we decompose the generalization error:

$$G(S, \theta) - G(S, \theta') = (R(S, \theta) - R(S, \theta')) + (\hat{R}(S, \theta') - \hat{R}(S, \theta)). \tag{19}$$

Then, we upper-bound the difference of risk terms:

$$R(S, \theta) - R(S, \theta') = \mathop{\mathbb{E}}_{z \sim \mathbb{D}} [L(A(S, \theta), z) - L(A(S, \theta'), z)] \leq \beta_{\Theta}. \tag{20}$$

Then, we upper-bound the difference of empirical risk terms:

$$\hat{R}(S, \theta') - \hat{R}(S, \theta) = \frac{1}{n} \sum_{i=1}^{n} L(A(S, \theta'), z_i) - L(A(S, \theta), z_i) \leq \beta_{\Theta}. \tag{21}$$

Combining Equations 19 to 21 completes the proof.  $\square$

Note that it is unnecessary to upper-bound the absolute difference in generalization error when using uniform stability, since it follows from the symmetry of the supremum over $S, S' \in \mathcal{Z}^n$ or $\theta, \theta' \in \Theta$.

## B.2 Proof of Theorem 1

PAC-Bayesian analysis typically requires a key step known as *change of measure*. For our first bound, we use a change of measure inequality based on the *Rényi divergence*,

$$D_\alpha(\mathbb{Q}\|\mathbb{P}) \triangleq \frac{1}{\alpha - 1} \ln \mathbb{E}_{X \sim \mathbb{P}} \left[ \left( \frac{\mathbb{Q}(X)}{\mathbb{P}(X)} \right)^\alpha \right].$$

**Lemma 6** ([1, Theorem 8]). *Let $X$ denote a random variable taking values in $\Omega$, and let $\varphi : \Omega \to \mathbb{R}$ denote a measurable function. Then, for any $\alpha > 1$, and any two distributions, $\mathbb{P}$ and $\mathbb{Q}$, on $\Omega$,*

$$\frac{\alpha}{\alpha - 1} \ln \mathbb{E}_{X \sim \mathbb{Q}} [\varphi(X)] \leq D_\alpha(\mathbb{Q}\|\mathbb{P}) + \ln \mathbb{E}_{X \sim \mathbb{P}} \left[ \varphi(X)^{\frac{\alpha}{\alpha - 1}} \right]. \tag{22}$$

An important special case of Lemma 6 is $\alpha = 2$, in which case

$$D_2(\mathbb{Q}\|\mathbb{P}) = \ln \mathbb{E}_{X \sim \mathbb{P}} \left[ \left( \frac{\mathbb{Q}(X)}{\mathbb{P}(X)} \right)^2 \right] = \ln \left( \chi^2(\mathbb{Q}\|\mathbb{P}) + 1 \right),$$

and, taking the exponent of Equation 22,

$$\mathbb{E}_{X \sim \mathbb{Q}} [\varphi(X)] \leq \sqrt{(\chi^2(\mathbb{Q}\|\mathbb{P}) + 1) \mathbb{E}_{X \sim \mathbb{P}} [\varphi(X)^2]}.$$

Thus, with $X \triangleq \theta$ and $\varphi(X) \triangleq G(S, \theta)$,

$$G(S, \mathbb{Q}) = \mathbb{E}_{\theta \sim \mathbb{Q}} [G(S, \theta)] \leq \sqrt{(\chi^2(\mathbb{Q}\|\mathbb{P}) + 1) \mathbb{E}_{\theta \sim \mathbb{P}} [G(S, \theta)^2]}.$$

Further, since $\mathbb{E}_{\theta \sim \mathbb{P}}[G(S, \theta)^2]$ is a nonnegative function of $S \sim \mathbb{D}^n$, Markov's inequality says that

$$\Pr_{S \sim \mathbb{D}^n} \left\{ \mathbb{E}_{\theta \sim \mathbb{P}} \left[ G(S, \theta)^2 \right] \geq \frac{1}{\delta} \mathbb{E}_{S \sim \mathbb{D}^n} \mathbb{E}_{\theta \sim \mathbb{P}} \left[ G(S, \theta)^2 \right] \right\} \leq \delta.$$

We therefore have that with probability at least $1 - \delta$ over draws of $S \sim \mathbb{D}^n$,

$$G(S, \mathbb{Q}) \leq \sqrt{(\chi^2(\mathbb{Q}\|\mathbb{P}) + 1) \frac{1}{\delta} \mathbb{E}_{S \sim \mathbb{D}^n} \mathbb{E}_{\theta \sim \mathbb{P}} [G(S, \theta)^2]}. \tag{23}$$

All that remains is to upper-bound $\mathbb{E}_{S \sim \mathbb{D}^n} \mathbb{E}_{\theta \sim \mathbb{P}} \left[ G(S, \theta)^2 \right]$, which can be accomplished via pointwise hypothesis stability.

**Lemma 7** ([3, Lemma 11]). *For any (randomized) learning algorithm, $A$, and $M$-bounded loss function, $L$,*

$$\mathbb{E}_{S \sim \mathbb{D}^n} \left[ G(S, \theta)^2 \right] \leq \frac{2M^2}{n} + \frac{12M}{n} \sum_{i=1}^{n} \mathbb{E}_{S \sim \mathbb{D}^n} \mathbb{E}_{z \sim \mathbb{D}} \left[ \left| L(A(S, \theta), z_i) - L(A(S^{i,z}, \theta), z_i) \right| \right]. \tag{24}$$

Taking the expectation over $\theta \sim \mathbb{P}$ on both sides of Equation 24, and using the linearity of expectation, we have that

$$\mathbb{E}_{S \sim \mathbb{D}^n} \mathbb{E}_{\theta \sim \mathbb{P}} \left[ G(S, \theta)^2 \right] \leq \frac{2M^2}{n} + \frac{12M}{n} \sum_{i=1}^{n} \mathbb{E}_{S \sim \mathbb{D}^n} \mathbb{E}_{z \sim \mathbb{D}} \mathbb{E}_{\theta \sim \mathbb{P}} \left[ \left| L(A(S, \theta), z_i) - L(A(S^{i,z}, \theta), z_i) \right| \right]$$

$$\leq \frac{2M^2}{n} + \frac{12M}{n} \sum_{i=1}^{n} \beta_{\mathcal{Z}} = \frac{2M^2}{n} + 12M\beta_{\mathcal{Z}}. \tag{25}$$

The last inequality follows directly from Definition 2. Combining Equations 23 and 25, we obtain Equation 5.

## B.3  Proof of Theorem 2

The proof of Theorem 2 requires two technical lemmas: the first is a change of measure inequality based on the KL divergence, attributed to Donsker and Varadhan [2]; the second is an upper bound on the moment-generating function of the generalization error, which we prove herein.

**Lemma 8** ([2]). *Let $X$ denote a random variable taking values in $\Omega$, and let $\varphi : \Omega \to \mathbb{R}$ denote a measurable function. Then, for any two distributions, $\mathbb{P}$ and $\mathbb{Q}$, on $\Omega$,*

$$\mathbb{E}_{X \sim \mathbb{Q}} [\varphi(X)] \leq D_{\mathrm{KL}}(\mathbb{Q} \| \mathbb{P}) + \ln \mathbb{E}_{X \sim \mathbb{P}} [\exp(\varphi(X))].$$

**Lemma 9.** *Fix a product measure, $\mathbb{P}$, on $\Theta = \prod_{t=1}^{T} \Theta_t$, and suppose $A$ is a $(\beta_{\mathcal{Z}}, \beta_{\Theta})$-uniformly stable with respect to $L$ and $\mathbb{P}$. Then, with*

$$\bar{\beta}_{\mathcal{Z}} = 2\beta_{\mathcal{Z}} + \frac{M}{n}, \tag{26}$$

*for any $\epsilon > 0$, the moment-generating function (MGF) of $G(S, \theta)$ satisfies*

$$\mathbb{E}_{S \sim \mathbb{D}^n} \mathbb{E}_{\theta \sim \mathbb{P}} [\exp(\epsilon\, G(S, \theta))] \leq \exp\left( \frac{\epsilon^2}{8} \left( n\bar{\beta}_{\mathcal{Z}}^2 + 4T\beta_{\Theta}^2 \right) + \epsilon\, \beta_{\mathcal{Z}} \right). \tag{27}$$

*Proof.* Before we begin, let us pause to recognize that the random variable $G(S, \theta)$ has nonzero mean. This is because the learning algorithm—hence, the loss composed with the learning algorithm—is a non-decomposable function of the training data and hyperparameters. We therefore start by defining a zero-mean random variable,

$$\Phi(S, \theta) \triangleq G(S, \theta) - G(\mathbb{D}, \mathbb{P}),$$

where

$$G(\mathbb{D}, \mathbb{P}) \triangleq \mathbb{E}_{S \sim \mathbb{D}^n} \mathbb{E}_{\theta \sim \mathbb{P}} [G(S, \theta)]$$

denotes the expected generalization error over draws of both $S \sim \mathbb{D}^n$ and $\theta \sim \mathbb{P}$. These definitions let us decompose the MGF of $G(S, \theta)$ as

$$\begin{aligned}
\mathbb{E}_{S \sim \mathbb{D}^n} \mathbb{E}_{\theta \sim \mathbb{P}} [\exp(\epsilon\, G(S, \theta))] &= \mathbb{E}_{S \sim \mathbb{D}^n} \mathbb{E}_{\theta \sim \mathbb{P}} [\exp(\epsilon\, \Phi(S, \theta) + \epsilon\, G(\mathbb{D}, \mathbb{P}))] \\
&= \mathbb{E}_{S \sim \mathbb{D}^n} \mathbb{E}_{\theta \sim \mathbb{P}} [\exp(\epsilon\, \Phi(S, \theta))] \exp(\epsilon\, G(\mathbb{D}, \mathbb{P})).
\end{aligned} \tag{28}$$

The second equality uses the fact that $G(\mathbb{D}, \mathbb{P})$ is constant with respect to the outer expectations. We now have that the MGF of $G(S, \theta)$ is the product of two factors: the MGF of $\Phi(S, \theta)$, and a monotonic function of $G(\mathbb{D}, \mathbb{P})$. We will bound these terms separately.

First, we upper-bound $G(\mathbb{D}, \mathbb{P})$ as follows:

$$\begin{aligned}
G(\mathbb{D}, \mathbb{P}) &= \mathbb{E}_{S \sim \mathbb{D}^n} \mathbb{E}_{\theta \sim \mathbb{P}} \left[ \mathbb{E}_{z \sim \mathbb{D}} [L(A(S, \theta), z)] - \frac{1}{n} \sum_{i=1}^{n} L(A(S, \theta), z_i) \right] \\
&= \frac{1}{n} \sum_{i=1}^{n} \mathbb{E}_{S \sim \mathbb{D}^n} \mathbb{E}_{z \sim \mathbb{D}} \mathbb{E}_{\theta \sim \mathbb{P}} [L(A(S, \theta), z) - L(A(S, \theta), z_i)] \\
&\leq \frac{1}{n} \sum_{i=1}^{n} \mathbb{E}_{S \sim \mathbb{D}^n} \mathbb{E}_{z \sim \mathbb{D}} \mathbb{E}_{\theta \sim \mathbb{P}} [L(A(S^{i,z}, \theta), z) - L(A(S, \theta), z_i)] + \beta_{\mathcal{Z}} \\
&= 0 + \beta_{\mathcal{Z}}.
\end{aligned}$$

In the second line, we rearrange the expectations using the linearity of expectation. In the third line, we form a new dataset, $S^{i,z}$, by replacing $z_i$ with $z$; via Definition 1, the expected difference of losses due to replacement, $\mathbb{E}_{\theta \sim \mathbb{P}}[L(A(S, \theta), z) - L(A(S^{i,z}, \theta), z)]$, is upper-bounded by $\beta_{\mathcal{Z}}$-uniform stability. The last line follows from the fact that each example is i.i.d.; since both $S$ and $S^{i,z}$ are distributed according to $\mathbb{D}^n$, and $\theta$ is independent of the datasets, the losses cancel out in expectation. Therefore, using the monotonicity of the exponent, and the fact that $\epsilon$ is positive, we have that

$$\exp(\epsilon\, G(\mathbb{D}, \mathbb{P})) \leq \exp(\epsilon\, \beta_{\mathcal{Z}}). \tag{29}$$

We now upper-bound the MGF of $\Phi(S,\theta)$, which involves a somewhat technical proof. To reduce notation, we omit the subscript on expectations. Further, we use the shorthand $z_{i:j} \triangleq (z_i, \ldots, z_j)$ and $\theta_{i:j} \triangleq (\theta_i, \ldots, \theta_j)$ to denote subsequences. (Interpret $z_{1:0}$ and $\theta_{1:0}$ as the empty set.) We start by constructing a Doob martingale as follows:

$$V_i \triangleq \begin{cases} \mathbb{E}[G(S,\theta) \,|\, z_{1:i}] - \mathbb{E}[G(S,\theta) \,|\, z_{1:i-1}] & \text{for } i = 1, \ldots, n; \\ \mathbb{E}[G(S,\theta) \,|\, S, \theta_{1:t}] - \mathbb{E}[G(S,\theta) \,|\, S, \theta_{1:t-1}] & \text{for } i = n+t, \ t = 1, \ldots, T. \end{cases}$$

Observe that $\mathbb{E}[V_i] = 0$ and $\sum_{i=1}^{n+T} V_i = \Phi(S,\theta)$. Thus, using the *law of total expectation* (alternatively, *law of iterated expectations*, or *tower rule*),

$$\mathbb{E}\left[\exp\left(\epsilon\,\Phi(S,\theta)\right)\right] \leq \left(\prod_{i=1}^{n} \sup_{z_{1:i-1}} \mathbb{E}\left[e^{\epsilon V_i} \,|\, z_{1:i-1}\right]\right)\left(\prod_{t=1}^{T} \sup_{S,\theta_{1:t-1}} \mathbb{E}\left[e^{\epsilon V_{n+t}} \,|\, S, \theta_{1:t-1}\right]\right). \quad (30)$$

Each iterate of Equation 30 is the supremum of the MGF for the corresponding martingale variable. We will use *Hoeffding's lemma* [6] to uniformly upper-bound each MGF. Hoeffding's lemma states that, if $X$ is a zero-mean random variable, such that $a \leq X \leq b$ almost surely, then for all $\epsilon \in \mathbb{R}$,

$$\mathbb{E}\left[e^{\epsilon X}\right] \leq \exp\left(\frac{\epsilon^2(b-a)^2}{8}\right). \quad (31)$$

To apply Hoeffding's lemma to each iterate of Equation 30, it suffices to show that

$$\forall i \in \{1, \ldots, n\},\ \exists c_i : \sup V_i - \inf V_i$$
$$= \sup_{\substack{z_{1:i}, z'_{1:i}: \\ z_{1:i-1}=z'_{1:i-1}}} \mathbb{E}[G(S,\theta) \,|\, z_{1:i}] - \mathbb{E}[G(S',\theta) \,|\, z'_{1:i}] \leq c_i; \quad (32)$$

$$\text{and}\quad \forall t \in \{1, \ldots, T\},\ \exists c_t : \sup V_{n+t} - \inf V_{n+t}$$
$$= \sup_S \sup_{\substack{\theta_{1:t}, \theta'_{1:t}: \\ \theta_{1:t-1}=\theta'_{1:t-1}}} \mathbb{E}[G(S,\theta) \,|\, S, \theta_{1:t}] - \mathbb{E}[G(S,\theta') \,|\, S, \theta'_{1:t}] \leq c_t. \quad (33)$$

The constants $c_i$ and $c_t$ replace $a - b$ in Equation 31.

To prove Equation 32, we use Lemma 4 (since $A$ is $\beta_{\mathcal{Z}}$-uniformly stable) and the independence of examples and hyperparameters. For any $z_{1:i}, z'_{1:i} \in \mathcal{Z}^i : z_{1:i-1} = z'_{1:i-1}$,

$$\mathbb{E}[G(S,\theta) \,|\, z_{1:i}] - \mathbb{E}[G(S',\theta) \,|\, z'_{1:i}] = \sum_{z_{i+1:n}} (G(S,\mathbb{P}) - G(S',\mathbb{P}))\prod_{j=1}^{n-i}\mathbb{D}(z_{i+j}) \leq \bar{\beta}_{\mathcal{Z}}.$$

(For notational simplicity, the expectation over $z_{i+1:n}$ is written as a summation, though $\mathcal{Z}$ need not be a finite set.) To prove Equation 33, we use Lemma 5 (since $A$ is $\beta_{\Theta}$-uniformly stable) and the independence of hyperparameters. For any $S \in \mathcal{Z}^n$ and $\theta_{1:t}, \theta'_{1:t} \in \prod_{j=1}^{t} \Theta_j : \theta_{1:t-1} = \theta'_{1:t-1}$,

$$\mathbb{E}[G(S,\theta) \,|\, S, \theta_{1:t}] - \mathbb{E}[G(S,\theta') \,|\, S, \theta'_{1:t}] = \sum_{\theta_{t+1:T}} (G(S,\theta) - G(S,\theta'))\prod_{j=1}^{T-t}\mathbb{P}(\theta_{t+j}) \leq 2\beta_{\Theta}.$$

Thus, applying Hoeffding's lemma (Equation 31) to each iterate of Equation 30—using $c_i = \bar{\beta}_{\mathcal{Z}}$ in Equation 32, and $c_t = 2\beta_{\Theta}$ in Equation 33—we have that

$$\mathbb{E}\left[\exp\left(\epsilon\,\Phi(S,\theta)\right)\right] \leq \left(\prod_{i=1}^{n}\exp\left(\frac{\epsilon^2\bar{\beta}_{\mathcal{Z}}^2}{8}\right)\right)\left(\prod_{t=1}^{T}\exp\left(\frac{\epsilon^2(2\beta_{\Theta})^2}{8}\right)\right)$$
$$= \exp\left(\frac{\epsilon^2}{8}\left(n\bar{\beta}_{\mathcal{Z}}^2 + 4T\beta_{\Theta}^2\right)\right). \quad (34)$$

Finally, by combining Equations 28, 29 and 34, we establish Equation 27. $\qquad \square$

We are now ready to prove Theorem 2. Let $\epsilon > 0$ denote a free parameter, which we will define later. Via Lemma 8 (with $X \triangleq \theta$ and $\varphi(X) \triangleq \epsilon\, G(S, \theta)$), we have that

$$G(S, \mathbb{Q}) = \frac{1}{\epsilon} \mathop{\mathbb{E}}_{\theta \sim \mathbb{Q}} [\epsilon\, G(S, \theta)] \leq \frac{1}{\epsilon} \left( D_{\mathrm{KL}}(\mathbb{Q}\|\mathbb{P}) + \ln \mathop{\mathbb{E}}_{\theta \sim \mathbb{P}} [\exp\left(\epsilon\, G(S, \theta)\right)] \right). \tag{35}$$

By Markov's inequality, with probability at least $1 - \delta$ over draws of $S \sim \mathbb{D}^n$,

$$\mathop{\mathbb{E}}_{\theta \sim \mathbb{P}} [\exp\left(\epsilon\, G(S, \theta)\right)] \leq \frac{1}{\delta} \mathop{\mathbb{E}}_{S \sim \mathbb{D}^n} \mathop{\mathbb{E}}_{\theta \sim \mathbb{P}} [\exp\left(\epsilon\, G(S, \theta)\right)]$$

$$\leq \frac{1}{\delta} \exp\left( \frac{\epsilon^2}{8} \left( n \bar{\beta}_{\mathcal{Z}}^2 + 4T\beta_{\Theta}^2 \right) + \epsilon \beta_{\mathcal{Z}} \right). \tag{36}$$

The second inequality uses Lemma 9 to upper-bound the MGF of $G(S, \theta)$, with $\bar{\beta}_{\mathcal{Z}}$ defined in Equation 26. Combining Equations 35 and 36, we thus have that with probability at least $1 - \delta$,

$$G(S, \mathbb{Q}) \leq \beta_{\mathcal{Z}} + \frac{1}{\epsilon} \left( D_{\mathrm{KL}}(\mathbb{Q}\|\mathbb{P}) + \ln \frac{1}{\delta} \right) + \frac{\epsilon}{8} \left( n \bar{\beta}_{\mathcal{Z}}^2 + 4T\beta_{\Theta}^2 \right). \tag{37}$$

What remains is to optimize $\epsilon$ to minimize the bound. Minimizing an expression of the form $a/\epsilon + b\epsilon$ is fairly straightforward; the optimal value for $\epsilon$ is $\sqrt{a/b}$. However, if we were to apply this formula to Equation 37, the optimal $\epsilon$ would depend on $\mathbb{Q}$ via the KL divergence term. Since we want the bound to hold simultaneously for all $\mathbb{Q}$, we need to define $\epsilon$ such that it does not depend on $\mathbb{Q}$. To do so, we construct an infinite sequence of $\epsilon$ values; for $i = 0, 1, 2, \dots$, let

$$\epsilon_i \triangleq 2^i \sqrt{\frac{8 \ln \frac{2}{\delta}}{n \bar{\beta}_{\mathcal{Z}}^2 + 4T\beta_{\Theta}^2}}. \tag{38}$$

For each $\epsilon_i$, we assign $\delta_i \triangleq \delta 2^{-(i+1)}$ mass to the probability that Equation 37 does not hold, substituting $\epsilon_i$ and $\delta_i$ for $\epsilon$ and $\delta$, respectively. Thus, by the union bound, with probability at least $1 - \sum_{i=0}^{\infty} \delta_i = 1 - \delta \sum_{i=0}^{\infty} 2^{-(i+1)} = 1 - \delta$, all $i = 0, 1, 2, \dots$ satisfy

$$G(S, \mathbb{Q}) \leq \beta_{\mathcal{Z}} + \frac{1}{\epsilon_i} \left( D_{\mathrm{KL}}(\mathbb{Q}\|\mathbb{P}) + \ln \frac{1}{\delta_i} \right) + \frac{\epsilon_i}{8} \left( n \bar{\beta}_{\mathcal{Z}}^2 + 4T\beta_{\Theta}^2 \right).$$

For any $\mathbb{Q}$, we select the optimal index, $i^\star$, as

$$i^\star = \left\lfloor \frac{1}{2 \ln 2} \ln \left( \frac{D_{\mathrm{KL}}(\mathbb{Q}\|\mathbb{P})}{\ln(2/\delta)} + 1 \right) \right\rfloor.$$

Since, with a bit of arithmetic,

$$\frac{1}{2} \sqrt{\frac{D_{\mathrm{KL}}(\mathbb{Q}\|\mathbb{P})}{\ln(2/\delta)} + 1} \leq 2^{i^\star} \leq \sqrt{\frac{D_{\mathrm{KL}}(\mathbb{Q}\|\mathbb{P})}{\ln(2/\delta)} + 1}, \tag{39}$$

combining Equations 38 and 39, we have that

$$\sqrt{\frac{2(D_{\mathrm{KL}}(\mathbb{Q}\|\mathbb{P}) + \ln \frac{2}{\delta})}{n \bar{\beta}_{\mathcal{Z}}^2 + 4T\beta_{\Theta}^2}} \leq \epsilon_{i^\star} \leq \sqrt{\frac{8(D_{\mathrm{KL}}(\mathbb{Q}\|\mathbb{P}) + \ln \frac{2}{\delta})}{n \bar{\beta}_{\mathcal{Z}}^2 + 4T\beta_{\Theta}^2}}.$$

It can also be shown [9] that

$$D_{\mathrm{KL}}(\mathbb{Q}\|\mathbb{P}) + \ln \frac{1}{\delta_{i^\star}} \leq \frac{3}{2} \left( D_{\mathrm{KL}}(\mathbb{Q}\|\mathbb{P}) + \ln \frac{2}{\delta} \right).$$

Therefore, with probability at least $1 - \delta$ over draws of $S \sim \mathbb{D}^n$, every posterior, $\mathbb{Q}$, satisfies

$$G(S, \mathbb{Q}) \leq \beta_{\mathcal{Z}} + \frac{1}{\epsilon_{i^\star}} \left( D_{\mathrm{KL}}(\mathbb{Q}\|\mathbb{P}) + \ln \frac{1}{\delta_{i^\star}} \right) + \frac{\epsilon_{i^\star}}{8} \left( n \bar{\beta}_{\mathcal{Z}}^2 + 4T\beta_{\Theta}^2 \right)$$

$$\leq \beta_{\mathcal{Z}} + \sqrt{\frac{n \bar{\beta}_{\mathcal{Z}}^2 + 4T\beta_{\Theta}^2}{2(D_{\mathrm{KL}}(\mathbb{Q}\|\mathbb{P}) + \ln \frac{2}{\delta})}} \cdot \frac{3}{2} \left( D_{\mathrm{KL}}(\mathbb{Q}\|\mathbb{P}) + \ln \frac{2}{\delta} \right)$$

$$+ \sqrt{\frac{8(D_{\mathrm{KL}}(\mathbb{Q}\|\mathbb{P}) + \ln \frac{2}{\delta})}{n \bar{\beta}_{\mathcal{Z}}^2 + 4T\beta_{\Theta}^2}} \cdot \frac{n \bar{\beta}_{\mathcal{Z}}^2 + 4T\beta_{\Theta}^2}{8}$$

$$= \beta_{\mathcal{Z}} + \sqrt{2 \left( D_{\mathrm{KL}}(\mathbb{Q}\|\mathbb{P}) + \ln \frac{2}{\delta} \right) \left( n \bar{\beta}_{\mathcal{Z}}^2 + 4T\beta_{\Theta}^2 \right)}.$$

Substituting Equation 26 for $\bar{\beta}_{\mathcal{Z}}$, we obtain Equation 6.

## B.4 Proof of Theorem 3

To accommodate all posteriors that might arise from drawing $S \sim \mathbb{D}^n$, it helps to consider $\mathbb{Q}$ as a function of $S$. Accordingly, we let $\mathbb{Q}_S$ denote the distribution induced by $S$. With $\delta_1 \triangleq \delta/2$, let

$$
E_1(S) \triangleq \mathbb{1}\left\{\exists \mathbb{Q} : G(S, \mathbb{Q}) \geq +\beta_{\mathcal{Z}} + \sqrt{2\left(D_{\mathrm{KL}}(\mathbb{Q}\|\mathbb{P}) + \ln\frac{2}{\delta_1}\right)\left(\frac{(M + 2n\beta_{\mathcal{Z}})^2}{n} + 4T\beta_{\Theta}^2\right)}\right\}
$$

denote the event that there exists a posterior for which Equation 6 does not hold. With $\delta_2 \triangleq \delta/2$, let

$$
E_2(S, \theta) \triangleq \mathbb{1}\left\{G(S, \theta) \geq G(S, \mathbb{Q}_S) + \beta_{\Theta}\sqrt{2\,T \ln\frac{1}{\delta_2}}\right\}
$$

denote the event that the generalization error for a given $\theta$ exceeds the expected generalization error under the posterior $\mathbb{Q}_S$ by more than $\beta_{\Theta}\sqrt{2\,T \ln\frac{1}{\delta_2}}$.

The probability we want to upper-bound is

$$
\Pr_{\substack{S \sim \mathbb{D}^n \\ \theta \sim \mathbb{Q}_S}}\{E_1(S) \vee E_2(S, \theta)\} \leq \Pr_{S \sim \mathbb{D}^n}\{E_1(S)\} + \Pr_{\substack{S \sim \mathbb{D}^n \\ \theta \sim \mathbb{Q}_S}}\{E_2(S, \theta)\}
$$

$$
\leq \Pr_{S \sim \mathbb{D}^n}\{E_1(S)\} + \sup_{S \in \mathcal{Z}^n}\Pr_{\theta \sim \mathbb{Q}_S}\{E_2(S, \theta)\,|\,S\}.
$$

The first inequality follows from the union bound; the second inequality follows from probability theory. By Theorem 2, $\Pr_{S \sim \mathbb{D}^n}\{E_1(S)\} \leq \delta_1$. To upper-bound $\Pr_{\theta \sim \mathbb{Q}_S}\{E_2(S, \theta)\,|\,S\}$, it suffices to show that $G(S, \theta)$ concentrates tightly around $G(S, \mathbb{Q}_S)$. We will do so with *McDiarmid's inequality* [10]. The following is a specialized version of the theorem.

**Lemma 10** ([10]). *Let $X_1, \ldots, X_n$ denote i.i.d. random variables, each taking values in $\Omega$. Suppose $\varphi : \Omega^n \to \mathbb{R}$ is a measurable function for which there exists a constant, $\beta$, such that*

$$
\sup_{\omega_1, \ldots, \omega_n \in \Omega^n} \sup_{\omega_i' \in \Omega} |\varphi(\omega_1, \ldots, \omega_i, \ldots, \omega_n) - \varphi(\omega_1, \ldots, \omega_i', \ldots, \omega_n)| \leq \beta. \tag{40}
$$

*Then, for any $\epsilon > 0$,*

$$
\Pr\{\varphi(X) - \mathbb{E}\,\varphi(X) \geq \epsilon\} \leq \exp\left(\frac{-2\epsilon^2}{n\beta^2}\right). \tag{41}
$$

An important special case is when $\beta = \Theta(n^{-1})$, in which case Equation 41 is $\mathrm{O}(\exp(-2n\epsilon^2))$, which decays rapidly.

Recall that $A$ is $\beta_{\Theta}$-uniformly stable with respect to $L$, independent of the posterior. Remember also that, by Lemma 5, $G$ satisfies McDiarmid's stability condition (Equation 40) with $\beta \triangleq 2\beta_{\Theta}$. Since $\mathbb{Q}_S$ is a product measure, we can therefore apply McDiarmid's inequality; with $\epsilon \triangleq \beta_{\Theta}\sqrt{2\,T \ln\frac{1}{\delta_2}}$,

$$
\Pr_{\theta \sim \mathbb{Q}_S}\{E_2(S, \theta)\,|\,S\} \leq \exp\left(\frac{-2\left(\beta_{\Theta}\sqrt{2\,T \ln\frac{1}{\delta_2}}\right)^2}{T\,(2\beta_{\Theta})^2}\right) = \delta_2.
$$

Thus,

$$
\Pr_{\substack{S \sim \mathbb{D}^n \\ \theta \sim \mathbb{Q}_S}}\{E_1(S) \vee E_2(S, \theta)\} \leq \delta_1 + \delta_2 = \delta;
$$

so, with probability at least $1 - \delta$,

$$
G(S, \theta) \leq \beta_{\Theta}\sqrt{2\,T \ln\frac{1}{\delta_2}} + G(S, \mathbb{Q}_S)
$$

$$
\leq \beta_{\Theta}\sqrt{2\,T \ln\frac{1}{\delta_2}} + \beta_{\mathcal{Z}} + \sqrt{2\left(D_{\mathrm{KL}}(\mathbb{Q}_S\|\mathbb{P}) + \ln\frac{2}{\delta_1}\right)\left(\frac{(M + 2n\beta_{\mathcal{Z}})^2}{n} + 4T\beta_{\Theta}^2\right)}.
$$

Replacing $\delta_1$ and $\delta_2$ with $\delta/2$ yields Equation 7.

## C   Efficient Iteratively Re-weighted Sampling

At each iteration of Algorithm 1, we sample from a categorical distribution on $\{1, \ldots, n\}$, then re-weight the distribution. While sampling from a uniform distribution is trivial, sampling from a nonuniform distribution is complicated. If the distribution is static, sampling can be performed in constant time, with $O(n)$ initialization time and $O(n)$ space, using the *alias method* [7]. However, the data structure that enables the alias method cannot be updated in sublinear time, which makes the alias method inefficient for iterative re-weighting when $n$ is large.

In this section, we describe an algorithm for iteratively re-weighted sampling that balances sampling efficiency with re-weighting efficiency. Like the alias method, the algorithm requires $O(n)$ initialization time and $O(n)$ space, but the cost of sampling and re-weighting is $O(\log n)$-time. Even for very large $n$, logarithmic time is an acceptable iteration complexity—especially since it may pale in comparison to the complexity of updating the hypothesis.

Before training, we initialize a full binary tree of depth $\lceil \log n \rceil$. We label the first $n$ leaves with the initial sampling weights (e.g., for uniform initialization, $n^{-1}$) and label the remaining $2^{\lceil \log n \rceil} - n$ leaves with 0. We then label each internal node with the sum of its children. During training, we sample from the distribution by performing a random tree traversal: at each internal node visited, we flip a biased coin, whose outcome probabilities are proportional to the labels of the node's children, then move to the corresponding child; the index of the leaf node we arrive at is the sampled value. It is easy to verify that this procedure results in a sample from the distribution. To modify the weight for a given index, we add the change in weight to each node in the path from the root to the associated leaf node. Pseudocode for these procedures is given in Algorithm 2.

---

**Algorithm 2** Efficient Iteratively Re-weighted Sampling

---

1: **procedure** INITIALIZE($q_1, \ldots, q_n$)
2:      Initialize a full binary tree, $\mathcal{T}$, of depth $\lceil \log n \rceil$
3:      For $i = 1, \ldots, n$, label the $i^{\text{th}}$ leaf node with $q_i$; label the remaining leaf nodes with 0
4:      Label each internal node with the sum of its children's labels.
5: **procedure** SAMPLE($\mathcal{T}$)
6:      $v \leftarrow$ ROOT($\mathcal{T}$)
7:      **while** $v$ is not a leaf **do**
8:          Flip a biased coin, $c$, with outcome probabilities proportional to the labels of $v$'s children
9:          **if** $c =$ HEADS **then**
10:              $v \leftarrow$ LEFTCHILD
11:          **else**
12:              $v \leftarrow$ RIGHTCHILD
13:      **return** index of leaf node $v$
14: **procedure** UPDATE($\mathcal{T}, i, q$)
15:      $\Delta \leftarrow q - q_i$
16:      **for** node $v$ on the path from the root to the $i^{\text{th}}$ leaf node **do**
17:          Add $\Delta$ to the label of $v$

---

## D   Proofs from Section 5

### D.1   Proof of Theorem 4

Observe that the KL divergence decomposes as

$$D_{\text{KL}}(\mathbb{Q}\|\mathbb{P}) = \mathop{\mathbb{E}}_{(i_1,\ldots,i_T)\sim\mathbb{Q}} \left[ \ln\left( \frac{\mathbb{Q}(i_1,\ldots,i_T)}{\mathbb{P}(i_1,\ldots,i_T)} \right) \right] = \sum_{t=1}^{T} \mathop{\mathbb{E}}_{(i_1,\ldots,i_t)\sim\mathbb{Q}} \left[ \ln\left( \frac{\mathbb{Q}_t(i_t)}{\mathbb{P}_t(i_t)} \right) \right], \qquad (42)$$

where $\mathbb{Q}_t(i) = \mathbb{Q}(i_t = i \,|\, i_1, \ldots, i_{t-1})$ is the conditional posterior at iteration $t$, and $\mathbb{P}_t(i)$, the conditional prior, is simply a uniform distribution on $\{1, \ldots, n\}$. In the first iteration, $\mathbb{Q}_1(i) = \mathbb{P}_1(i)$, since the sampling weights are initialized uniformly to 1. Then, for every $t \geq 2$,

$$\ln\left( \frac{\mathbb{Q}_t(i_t)}{\mathbb{P}_t(i_t)} \right) = \ln\left( \frac{q_{i_t}^{(t)} / \sum_{i=1}^{n} q_i^{(t)}}{n^{-1}} \right) = \ln q_{i_t}^{(t)} - \ln\left( \frac{1}{n}\sum_{i=1}^{n} q_i^{(t)} \right), \qquad (43)$$

where $q_i^{(t)}$ denotes the state of $q_i$ at the *start* of the $t^{\text{th}}$ iteration. Unrolling the recursive definition of $q_i^{(t)}$, we have

$$\ln q_i^{(t)} = \ln \prod_{j=1}^{N_{i,t}} \exp\left(\alpha\, f(z_i, h_{O_{i,j}})\, \tau^{N_{i,t}-j}\right) = \alpha \sum_{j=1}^{N_{i,t}} f(z_i, h_{O_{i,j}})\, \tau^{N_{i,t}-j}. \tag{44}$$

Further, using Jensen's inequality and the concavity of the logarithm,

$$
\begin{aligned}
\ln\left(\frac{1}{n}\sum_{i=1}^{n} q_i^{(t)}\right) &= \ln\left(\frac{1}{n}\sum_{i=1}^{n}\prod_{j=1}^{N_{i,t}} \exp\left(\alpha\, f(z_i, h_{O_{i,j}})\, \tau^{N_{i,t}-j}\right)\right)\\
&\geq \frac{1}{n}\sum_{i=1}^{n}\ln\left(\prod_{j=1}^{N_{i,t}} \exp\left(\alpha\, f(z_i, h_{O_{i,j}})\, \tau^{N_{i,t}-j}\right)\right)\\
&= \frac{\alpha}{n}\sum_{i=1}^{n}\sum_{j=1}^{N_{i,t}} f(z_i, h_{O_{i,j}})\, \tau^{N_{i,t}-j}.
\end{aligned}
\tag{45}
$$

Combining Equations 42 to 45, we have

$$D_{\mathrm{KL}}(\mathbb{Q}\|\mathbb{P}) \leq \sum_{t=2}^{T}\ \mathop{\mathbb{E}}_{(i_1,\dots,i_t)\sim\mathbb{Q}}\left[\alpha\sum_{j=1}^{N_{i_t,t}} f(z_{i_t}, h_{O_{i_t,j}})\, \tau^{N_{i_t,t}-j} - \frac{\alpha}{n}\sum_{i=1}^{n}\sum_{k=1}^{N_{i,t}} f(z_i, h_{O_{i,k}})\, \tau^{N_{i,t}-k}\right].$$

We then reorder the summations to obtain Equation 9.

## D.2  Proof of Theorem 5

First, observe that the lower bound in Equation 45 is nonnegative, due to the nonnegativity of the utility function, amplitude and decay. We can therefore drop $\ln\left(\frac{1}{n}\sum_{i=1}^{n} q_i^{(t)}\right)$ from Equation 43, which yields the following upper bound:

$$D_{\mathrm{KL}}(\mathbb{Q}\|\mathbb{P}) \leq \mathop{\mathbb{E}}_{(i_1,\dots,i_T)\sim\mathbb{Q}}\left[\sum_{t=2}^{T}\ln q_{i_t}^{(t)}\right] = \mathop{\mathbb{E}}_{(i_1,\dots,i_T)\sim\mathbb{Q}}\left[\alpha\sum_{t=2}^{T}\sum_{j=1}^{N_{i_t,t}} f(z_{i_t}, h_{O_{i_t,j}})\, \tau^{N_{i_t,t}-j}\right]. \tag{46}$$

Since $i_t = i_{O_{i_t,j}}$ for all $j \in N_{i_t,t}$, we have that

$$f(z_{i_t}, h_{O_{i_t,j}}) = f(z_{i_{O_{i_t,j}}}, h_{O_{i_t,j}}) = f(z_{i_{t'}}, h_{t'})$$

for every $t' < t : i_t = i_{t'}$. Thus, the $t^{\text{th}}$ computed utility value, $f(z_{i_t}, h_t)$, is referenced whenever the same index is sampled in future iterations. We can therefore reorder the above summations as

$$\sum_{t=2}^{T}\sum_{j=1}^{N_{i_t,t}} f(z_{i_t}, h_{O_{i_t,j}})\, \tau^{N_{i_t,t}-j} = \sum_{t=1}^{T-1} f(z_{i_t}, h_t)\sum_{j=1}^{N_{i_t,T+1}-N_{i_t,t+1}} \tau^{j-1}. \tag{47}$$

Note that when $N_{i_t,T+1} - N_{i_t,t+1} = 0$ (i.e., when $i_t$ is not sampled again in iterations $t+1,\dots,T$), the inner summation evaluates to zero. Since the utility function and amplitude are nonnegative, adding a term for $i_t$ that never appears again can only increase the bound. Thus, we can simplify the above expression by extending the inner summation to an infinite series:

$$\sum_{j=1}^{N_{i_t,T+1}-N_{i_t,t+1}} \tau^{j-1} \leq \sum_{j=0}^{\infty} \tau^{j} \leq \frac{1}{1-\tau}. \tag{48}$$

The last inequality follows from the geometric series identity, since $\tau \in (0,1)$. Combining Equations 46 to 48 yields Equation 10.