[Reviews · NeurIPS 2017]

Reviewer 1



This work analyzes the stability of SGD based learning algorithms. Both the stability of the algorithm with respect to small changes on the training set and small changes on the sequence of samples are analyzed. Those two terms take place in a PAC Bayes bound which suggests that sampling from a posterior distribution over the training set can be more efficient than sampling from a uniform distribution. On the light of this result, they develop Algorithm1, which dynamically produces a posterior focusing on harder examples during training. Experimental results on CIFAR-10 shows faster convergence rate. I did not verified the proofs, but the paper is well written and sound. The end algorithm is easy to implement and can be added to many deep learning algorithms with no effort. The end algorithm is analogous to active learning and the proposed utility function may not work well for some dataset. For example some sample might have high aleatoric uncertainty and the current algorithm would mainly be “obsessed” by unsolvable examples. Using a utility function based on epistemic uncertainty instead would solve this issue. (See Yarin Gal’s work on active learning). I believe that this algorithm would be good at solving the class imbalance issue, e.g., class A has 10000 samples and class B has 100 samples. Having an experiment highlighting this would add great value to this work (this is not a request, just a suggestion) Minors: * Line 88: the indicator function should have \neq inside and not “=“. * Line 39: The space of X and the space of H both uses R^d and they are not the same space * Line 50: Typo as a a.

Reviewer 2



The paper is interesting and definitely of interest to the NIPS community. In particular, the idea of coupling PAC-Bayes and stability is appealing. Overall, I stand for acceptance, although I have some remarks: * Lines 69-81: I am afraid I have not understood the message here. Too subtle indeed. * The authors assume that the loss is bounded. Could this stringent assumption be relaxed? Some effort has been put lately to derive PAC-Bayesian bounds for unbounded losses and/or heavy-tailed distributions, see for example Catoni (2004, St Flour lecture notes), Audibert and Catoni (2011, Annals of Statistics), Grünwald and Mehta (2016, arXiv preprint), Alquier and Guedj (2016, arXiv preprint). * Most serious concern: in Th. 2 and 3, do the risk bounds hold for the SGD (alg 1)? If not, what is $T$ standing for? This section of the paper is not clear enough. I feel that a very specific risk bound for $\beta$-{uniformly,hypothesis}-stable SGD with the actual $\beta$ terms would be much more clearer to readers. * Th4: to my opinion this result is the most interesting of the paper but curiously this is not put forward. Could a similar result be derived for non-uniform priors (e.g., enforcing sparsity)? * Some typos (lines 110, 170).

Reviewer 3



The paper opens the way to a new use of PAC-Bayesian theory, by combining PAC-Bayes with algorithmic stability to study stochastic optimization algorithms. The obtained probabilistic bounds are then used to inspire adaptive sampling strategies, studied empirically in a deep learning scenario. The paper is well written, and the proofs are non-trivial. It contains several clever ideas, namely the use of algorithmic stability to bound the complexity term inside PAC-Bayesian bounds. It's also fruitful to express the prior and posterior distributions over the sequences of indexes used by a stochastic gradient descent algorithm. Up to my knowledge, this is very different than any other previous PAC-Bayes theorems, and I think it can inspire others in the future. The experiments using a (moderately) deep network, trained by both SGD and AdaGrad, are convincing enough; they show that the proposed adaptive sampling strategy can benefit to existing optimization methods simply by selecting the samples during training time. Small comments: - Line 178: In order to sell their result, the authors claim that "high-probability bounds are usually favored" over bounds in expectation. I don't think the community is unanimous about this, and I would like the authors to convince me that I should prefer high-probability bounds. - Section 5 and Algorithm 1: I suggest to explicitly express the utility function f as a function of a model h. Typos: - Line 110: van -> can - Supplemental, Line 183: he -> the ** UPDATE ** I encourage the authors to carefully consider the reviewer's comments while preparing the final version of their paper. Concerning the claim that high-probability bound imply expectation bound, I think it's right, but it deserves to be explained properly to convince the readers.